# Prolonged nicotine exposure reduces aversion to the drug in mice by altering nicotinic transmission in the interpeduncular nucleus

Sarah Mondoloni[1], Claire Nguyen[1], Eléonore Vicq[1,2], Maria Ciscato[2], Joachim Jehl[1,2], Romain Durand-de Cuttoli[1], Nicolas Torquet[1], Stefania Tolu[1], Stéphanie Pons[3], Uwe Maskos[3], Fabio Marti[1,2], Philippe Faure[1,2]*[†], Alexandre Mourot[1,2]*[†]

[1]Sorbonne Université, Inserm, CNRS, Neuroscience Paris Seine - Institut de Biologie Paris Seine (NPS - IBPS), Paris, France; [2]Brain Plasticity Unit, CNRS, ESPCI Paris, PSL Research University, Paris, France; [3]Institut Pasteur, Unité Neurobiologie intégrative des systèmes cholinergiques, Département de neuroscience, Paris, France

*For correspondence:
phfaure@gmail.com (PF);
almourot@gmail.com (AM)

†These authors contributed equally to this work

Competing interest: The authors declare that no competing interests exist.

**Abstract** Nicotine intake is likely to result from a balance between the rewarding and aversive properties of the drug, yet the individual differences in neural activity that control aversion to nicotine and their adaptation during the addiction process remain largely unknown. Using a two-bottle choice experiment, we observed considerable heterogeneity in nicotine-drinking profiles in isogenic adult male mice, with about half of the mice persisting in nicotine consumption even at high concentrations, whereas the other half stopped consuming. We found that nicotine intake was negatively correlated with nicotine-evoked currents in the interpeduncular nucleus (IPN), and that prolonged exposure to nicotine, by weakening this response, decreased aversion to the drug, and hence boosted consumption. Lastly, using knock-out mice and local gene re-expression, we identified β4-containing nicotinic acetylcholine receptors of IPN neurons as molecular and cellular correlates of nicotine aversion. Collectively, our results identify the IPN as a substrate for individual variabilities and adaptations in nicotine consumption.

## Editor's evaluation

The vulnerability to adaptations in nicotine addiction is largely determined by individual differences in neural activity controlling nicotine aversion. In the current study, Mondoloni and colleagues differentiated individual mice into "avoiders" and "non-avoiders" based on their nicotine drinking behavior in two-bottle choice tests, and identified a nicotinic receptor, β4 nAChR, in the interpeduncular nuclei as a key substrate for mediating nicotine aversion. This finding has important implications for understanding individual differences in drug addiction.

## Introduction

Nicotine remains one of the most-widely used addictive substance in the world, and even though cigarette smoking is overall decreasing, the use of new products such as electronic cigarettes has risen dramatically in recent years (*WHO, 2021*). Nicotine administration induces a range of effects, from pleasant (i.e. appetitive, rewarding, reinforcing, anxiolytic…) to noxious (i.e. anxiogenic, aversive…) (*Fowler and Kenny, 2014*; *Verendeev and Riley, 2013*; *Wills et al., 2022*). These multifaceted

effects have been described both in humans and in rodents. They greatly depend on the dose of nicotine administered, show substantial inter-individual variability, and are considered essential in the regulation of nicotine intake and in the maintenance of addiction. Understanding the variable effects of nicotine at the molecular and circuit levels is therefore fundamental to progress in the pathophysiology of nicotine addiction and to develop efficient smoking-cessation therapies.

Nicotine mediates its physiological effects by activating nicotinic acetylcholine receptors (nAChRs), pentameric ligand-gated ion channels encoded by a large multigene family (*Wills et al., 2022*). There are nine α (α2–10) and three β (β2–4) nAChR subunits expressed in the brain, which can assemble to form homo-pentamers or hetero-pentamers with various localizations and functions (*Taly et al., 2009*; *Zoli et al., 2015*). Initiation of consumption and reinforcement to nicotine involve the mesolimbic dopamine reward circuit, which originates in the ventral tegmental area (VTA; *Maskos et al., 2005*). Nicotine primarily acts on this circuit by activating α4β2 nAChRs, a receptor subtype that displays high affinity for the drug (*Durand-de Cuttoli et al., 2018*; *Maskos et al., 2005*; *Tapper et al., 2004*; *Tolu et al., 2013*). An acute injection of nicotine also inhibits a subset of VTA dopamine neurons that project to the amygdala (*Nguyen et al., 2021*), and this results in elevated anxiety in mice, illustrating the heterogeneity of the brain reward circuit, and the complexity of nicotine dependence.

Another important pathway in the neurobiology of nicotine addiction is the medial habenulo-interpeduncular (MHb-IPN) axis (*Fowler et al., 2011*; *Frahm et al., 2011*; *Molas et al., 2017*; *Morton et al., 2018*; *Tuesta et al., 2017*; *Wolfman et al., 2018*). This pathway is deeply implicated in the regulation of aversive physiological states such as fear and anxiety (*Molas et al., 2017*; *Otsu et al., 2019*; *Yamaguchi et al., 2013*; *Zhang et al., 2016*). It is believed to directly mediate aversion to high doses of nicotine (*Fowler et al., 2011*; *Frahm et al., 2011*), to trigger affective (anxiety) and somatic symptoms following nicotine withdrawal (*Pang et al., 2016*; *Salas et al., 2009*; *Zhao-Shea et al., 2015*; *Zhao-Shea et al., 2013*) and to be involved in relapse to nicotine-seeking (*Forget et al., 2018*). Strikingly, neurons of the MHb-IPN axis express the highest density and largest diversity of nAChRs in the brain, notably the rare α5, α3, and β4 subunits (*Zoli et al., 2015*). These are encoded by the *CHRNA5-A3-B4* gene cluster, some sequence variants of which are associated with a high risk of addiction in humans (*Bierut et al., 2008*; *Lassi et al., 2016*). The α3 and β4 subunits are virtually absent in the VTA, or in other parts of the brain. The implication of these different subunits in the response of the IPN to nicotine was so far investigated using brain slice physiology experiments or indirect approaches such as c-fos quantification (*Fowler et al., 2011*; *Wolfman et al., 2018*). Recording the physiological response of IPN neurons to nicotine in the intact brain in vivo remains a prerequisite to understand the mechanism of action of nicotine on this pathway.

A distinct feature of addiction is that, overall, only some individuals lose control over their drug use, progressively shifting to compulsive drug intake (*Deroche-Gamonet et al., 2004*; *George and Koob, 2017*; *Juarez et al., 2017*; *Pascoli et al., 2018*; *Siciliano et al., 2019*). About one-third to one-half of people who have tried smoking tobacco become regular users (*Centers for Disease Control and Prevention, 2010*). Individual differences in the sensitivities of the VTA and MHb-IPN systems, and in their respective adaptations during chronic tobacco use, could contribute to the vulnerability to nicotine and to the severity of the addiction process. Yet, the neural mechanism that makes individuals more prone to maintain nicotine consumption than durably stop are unclear. In addition, whether the MHb-IPN pathway respond differently to nicotine in individuals with and without a history of nicotine use, and the mechanisms by which smokers develop tolerance to the aversive effects of nicotine, are largely unknown. Here, we used isogenic mice and electrophysiology (ex vivo and in vivo) to study the neuronal correlates of inter-individual variabilities in nicotine consumption behavior, and their adaptation after chronic exposure to the drug.

## Results

### Heterogeneity in nicotine consumption in isogenic wild-type mice

We used a continuous access, two-bottle choice nicotine-drinking test to assess consumption profiles in male wild-type (WT) C57BL/6 mice single-housed in their home cage (*Figure 1A*). In this test, animals have continuous and concurrent access to two bottles containing a solution of either 2% saccharine (vehicle) or nicotine plus 2% saccharine (to mask the bitter taste of nicotine). After a 4-day habituation period with water in both bottles, nicotine concentration was progressively increased

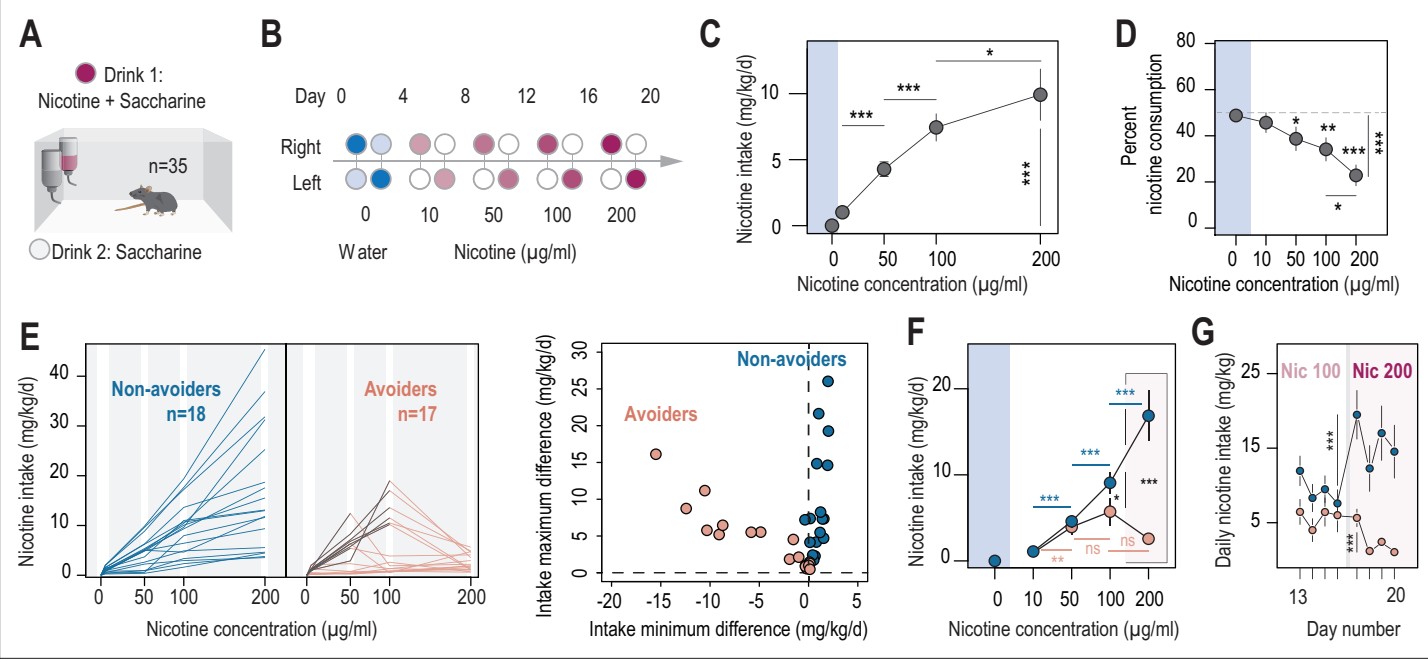

**Figure 1.** Two different profiles, avoiders and non-avoiders, emerged in WT mice subjected to a two-bottle choice nicotine-drinking test. (**A**) Continuous access, two-bottle choice setup. (**B**) Two-bottle choice paradigm. Each dot represents a bottle and is color-coded according to whether it contains water (blue or light blue), nicotine plus 2% saccharine (red, gradient of color intensities according to the nicotine concentration), or 2% saccharine (white) solutions. The nicotine concentration in the bottle increased progressively from 10 to 50, 100 and 200 µg/ml. Each condition lasted 4 days, and the bottles were swapped every other day. (**C**) Nicotine intake (mg/kg/day), averaged over 4 days, at different nicotine concentrations (Friedman test, n=35, df = 3, p<0.001 and Mann-Whitney post-hoc test with Holm-Bonferroni correction). (**D**) Percent nicotine consumption in WT mice for each concentration of nicotine, averaged over 4 days (Friedman test, n=35, df = 4, p<0.001 and Mann-Whitney post-hoc test with Holm-Bonferroni correction). (**E**) Left, nicotine intake in individual avoiders (n=17) and non-avoiders (n=18). Right, minimum and maximum values of the difference in nicotine intake between 2 consecutive days, for each individual. (**F**) Nicotine intake in avoiders and non-avoiders for each nicotine concentration, averaged over 4 days (Mann-Whitney comparison with a Holm-Bonferroni correction). (**G**) Daily nicotine intake in avoiders and non-avoiders for nicotine 100 and 200 µg/ml (paired Mann-Whitney). Note the drop in nicotine consumption at day 18 for avoiders. In all figure panels, avoiders are depicted in pinkish-orange while non-avoiders are in blue. *** p<0.001, ** p<0.01, * p<0.05.

The online version of this article includes the following source data and figure supplement(s) for figure 1:

**Source data 1.** Source data for *Figure 1*.

**Figure supplement 1.** Nicotine intake and consumption profiles in avoiders and non-avoiders.

in one bottle across 16 days, from 10 to 200 µg/ml (4 days at each concentration), while alternating the side of the nicotine-containing solution every other day to control for side bias (*Figure 1B*). The consumption from each bottle was measured every minute. We found that daily nicotine intake increased throughout the paradigm, stabilizing at about 10 mg/kg/day on average for the highest nicotine concentration tested (*Figure 1C*). Overall, the percent of nicotine consumption, that is, the nicotine solution intake relative to the total fluid intake, was initially close to 50%, and decreased for nicotine concentrations above 50 µg/ml (*Figure 1D*). These results match what was observed in previous studies using male C57BL/6 mice, notably that these mice rarely show nicotine consumption over 50%, whether the water is supplemented with saccharine or not (*Bagdas et al., 2019*; *Matta et al., 2006*). The decrease in percent nicotine consumption observed at the population level over the course of the task suggests that mice adapt their behavior to reduce their number of visits to the nicotine-containing bottle. Indeed, we observed that mice responded rapidly (within a day) to the increase in nicotine concentration by adjusting their nicotine intake (*Figure 1—figure supplement 1A*), resulting in titration of the nicotine dose, as previously reported (*Fowler et al., 2011*; *Tuesta et al., 2017*).

We noticed some disparity between mice and decided to analyze nicotine consumption profiles for each individual more closely. In particular, some mice abruptly reduced their intake right after an increase in nicotine concentration in the bottle, or had very low nicotine intake throughout the task

(intake never exceeded 2 mg/kg/day; *Figure 1E*). We classified these mice (17/35) as 'avoiders'. The other half of the mice (18/35), on the other hand, displayed a continuous increase in nicotine intake, or eventually reached a titration plateau in their consumption. These mice were classified as 'non-avoiders'. Another way to examine these distinct consumption profiles is to quantify the differences in intake between 2 consecutive days (positive differences indicate an increase in intake, while negative differences indicate a decrease in intake). The distinction between the two phenotypic groups becomes clear when we plot, for each mouse, the minimum and maximum values of the intake difference between 2 consecutive days (*Figure 1E*, right). The group of avoiders was characterized either by a negative minimum difference in intake (mice that reduce their intake abruptly), or by a minimum and maximum difference in intake close to zero (very low intake throughout the task). In contrast, the non-avoider mice were characterized by a minimum difference in intake between 2 consecutive days that was always positive, indicating a continuous increase in intake throughout the task. Overall, only a small proportion of the mice (7/35, all non-avoiders) reached a plateau in their consumption, which somewhat contrasts with the apparent titration observed at the population level, either here (*Figure 1C*) or in previous studies (*Fowler et al., 2011*; *Tuesta et al., 2017*). Avoiders and non-avoiders (phenotypes defined as above throughout the manuscript) showed on average similar nicotine intake for low concentrations of nicotine (10 and 50 μg/ml, p>0.05), but while nicotine intake increased throughout the task for non-avoiders (16.9±2.9 mg/kg/day for 200 μg/ml of nicotine) it dropped down to 2.6±0.4 mg/kg/day for such high nicotine concentration in avoiders (*Figure 1F*), and almost reached zero over the last 3 days (*Figure 1G*). The percent nicotine consumption was fairly constant throughout the task in non-avoiders, whereas in avoiders, it drastically decreased as nicotine concentration increased, to approximate zero at the end of the task (*Figure 1—figure supplement 1B*). We then compared the level of aversion produced by nicotine in avoiders and non-avoiders, with that produced by quinine, a notoriously bitter molecule. We found that all naive mice actively avoided the quinine-containing solution, and showed near-zero percent quinine consumption, as was observed with nicotine for avoiders, but not for non-avoiders (*Figure 1—figure supplement 1C*). We then used lower concentrations of quinine, and found that nicotine avoidance was not correlated with quinine avoidance (*Figure 1—figure supplement 1D*). Taken together, these results indicate that individuals who avoid nicotine have a strong aversion to the drug that is not directly tied to their sensitivity to the bitter taste of nicotine.

## The concentration of nicotine that triggers aversion differs among mice

Owing to the oral nature of the test, it may be difficult for mice to associate consumption in a particular bottle with the physiological effects (positive or negative) of nicotine. We thus systematically analyzed the percent nicotine consumption throughout the task in individual mice, to examine variability in their choice patterns and behavioral alterations. We found that some non-avoiders actively tracked the side associated with nicotine when the bottles were swapped (e.g. mouse #1 in *Figure 2A*), indicating a strong preference for the nicotine-containing bottle and active consumption. Other non-avoiders displayed a strong side preference and never alternated drinking side (e.g. mouse #2 in *Figure 2A*), and hence consumed nicotine in a more passive fashion. In contrast, all avoider mice (n=17) displayed active avoidance of the nicotine-containing solution, whether they initially tracked the nicotine solution (e.g. mice #3 and 4 in *Figure 2A*) or not (e.g. mice #5 and 6 in *Figure 2A*). To quantify the evolution of nicotine consumption at the individual level throughout the task, and to better account for the passive consumption behavior of some of the mice, we mapped each profile in a pseudo-ternary plot where the base represents the nicotine consumption index (from 0 to 100%), while the upper apex represents 100% side bias (*Figure 2B*). Such a ternary representation enables us to graphically distinguish between mice that actively track the nicotine bottle (bottom right apex, 100% nicotine consumption index), mice that actively avoid nicotine (bottom left apex, 0% nicotine consumption index), and mice that have a strong side bias (top apex), and to calculate the shortest distance for each mouse to each of the three apices. In addition, by representing the trajectory for each individual from the water condition to the 200 μg/ml nicotine condition, this graph can be used to reveal and quantify behavioral adaptations (or lack thereof) in each individual. Overall, we found that the behavior of non-avoiders was on average fairly consistent throughout the task, that is their distance from the three apices was not really impacted by the modifications in nicotine concentration (*Figure 2C* and *Figure 2—figure supplement 1A and B*), whether they consumed nicotine actively

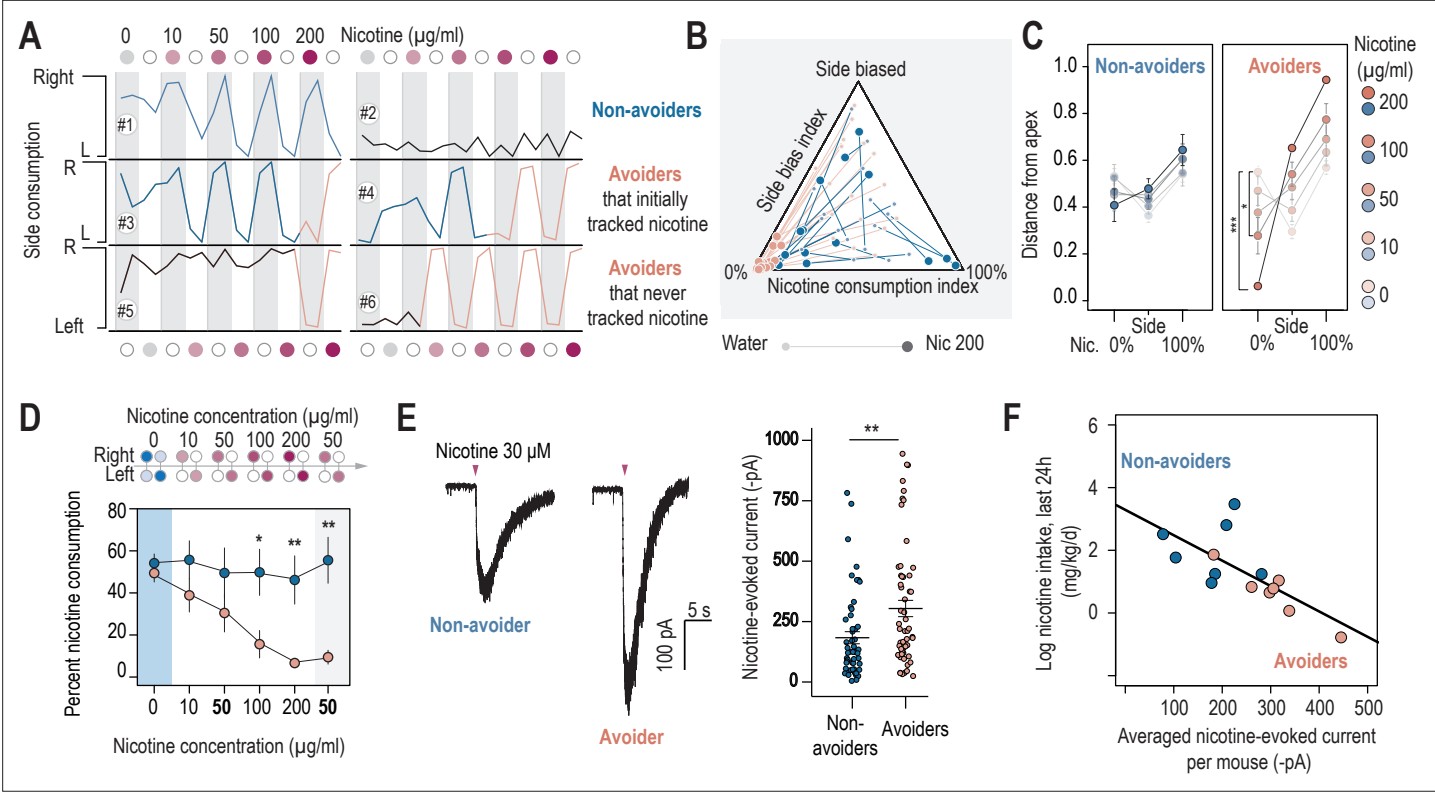

**Figure 2.** Nicotine intake was negatively correlated with the amplitude of the response to nicotine in IPN neurons. (**A**) Representative examples of choice behaviors (% consumption on the right vs. left bottle) in WT mice when the right-hand side bottle contains either nicotine +saccharine (red dots, grey stripe) or saccharine only (white dots, white stripes). Mice are same as in **Figure 1**. (**B**) Pseudo-ternary diagram representing, for each individual (18 non-avoiders and 17 avoiders, see **Figure 1**), its nicotine consumption index over its side bias index. Bottom left apex: 0% nicotine consumption (0%); Bottom right apex: 100% nicotine consumption (100%); Top apex: 100% side preference (Side biased, i.e. mice that never switch side). Small dots correspond to the habituation period (water vs. water) while bigger dots correspond to the condition with 200 μg/ml of nicotine in one bottle. Note how all avoider mice end up in the bottom left apex (0% nicotine consumption) at the end of the task. (**C**) Average distance from the three apices for each condition in the task (0, 10, 50, 100, and 200 μg/ml of nicotine, color-coded from light to dark), for avoiders and non-avoiders. Only avoider mice significantly changed their drinking strategy as nicotine concentration increased (paired Mann-Whitney test with Holm-Bonferroni correction). (**D**) Average percent nicotine consumption for avoider (n=8 mice) and non-avoider mice (n=8) for each concentration of nicotine. The 50 μg/ml nicotine solution was presented a second time to the mice, at the end of the session, for 4 days. (Mann-Whitney test, Holm-Bonferroni correction, $p_{(50 Nic)}$=0.04, $p_{(100 Nic)}$=0.005, $p_{(200 Nic)}$=0.004). (**E**) Representative (left) and average (right) currents recorded in voltage-clamp mode (–60 mV) from IPN neurons of non-avoiders (blue, n=52 neurons from 7 mice, I=–183.20 ± 25 pA) and avoiders (red, n=57 neurons from 7 mice, I=–254.64 ± 33 pA) following a puff application of nicotine (30 μM, 200ms). Avoiders presented greater nicotine-evoked currents than non-avoiders (Mann-Whitney test, p=0.0027). (**F**) Correlation between the dose consumed (log scale, over the last 24 hr prior to the recording) and the averaged nicotine evoked-current (-pA) per mouse (n=14 mice, $R^2$=0.47, $F_{1-12}$ = 12.45, p=0.004). In all figure panels, avoiders are depicted in pinkish-orange and non-avoiders in blue. *** p<0.001, ** p<0.01, * p<0.05.

The online version of this article includes the following source data and figure supplement(s) for figure 2:

**Source data 1.** Source data for **Figure 2**.

**Figure supplement 1.** Behavioral and electrophysiological characterization of avoiders and non-avoiders.

or passively. In contrast, the behavior of avoiders was highly nicotine concentration-dependent, with mice going further away from the 100% nicotine apex as nicotine concentration increases (**Figure 2C** and **Figure 2—figure supplement 1A, B**). Aversion to nicotine in avoider mice mainly occurred at the transition from 100 to 200 μg/ml, but some mice displayed aversion at concentrations as low as 10 μg/ml (**Figures 1E, 2A and C**). Together, these results show that most mice can actively track or avoid nicotine, indicating that they can discriminate nicotine from the control solution. Avoiders started actively avoiding the nicotine-containing bottle once a specific drug concentration was reached, suggesting the existence of a threshold at which nicotine aversion is triggered, which apparently differs between mice.

## Persistence of nicotine aversion

Do avoiders learn to stay away from the nicotine-containing solution, or do they just rapidly react to the change in nicotine concentration in the bottle to adjust their daily intake? To answer this question, we added at the end of the two-bottle choice task, that is after the 200 μg/ml nicotine concentration, a condition with a low concentration of nicotine (50 μg/ml) for 4 days. We chose 50 μg/ml of nicotine because avoiders and non-avoiders initially displayed comparable nicotine intake (*Figure 1F*) and percent consumption (*Figure 1—figure supplement 1B*) at this concentration. We hypothesized that if avoiders increased their percent nicotine consumption at the 200–50 μg/ml transition, this would indicate a rapid adjustment to the concentration proposed, in order to maintain their level of intake constant. In contrast, if avoiders maintained a steady, low percent nicotine consumption, this would indicate that nicotine aversion persists, regardless of the dose. Indeed, we found that lowering nicotine concentration from 200 to 50 μg/ml did not increase percent nicotine consumption in avoiders (*Figure 2D*), at least for the 4 days that mice were subjected to this concentration. We then performed a complementary experiment, in which mice were directly subjected to a high concentration of nicotine (200 μg/ml), followed by 8 days at 50 μg/ml. We found that, overall, mice avoided the 200 μg/ml nicotine solution, and that the following increase in nicotine preference was gradual at 50 μg/ml (*Figure 2—figure supplement 1C*). This slow adjustment to a lower-dose contrasts with the rapid (within a day) change in intake observed when nicotine concentration increases (see for instance *Figure 1—figure supplement 1A*). If we look at individuals, we observe that half of the mice (6/13) retained a steady, low nicotine preference (<20%) throughout the 8 days at 50 μg/ml (*Figure 2—figure supplement 1C*), which is similar to what was observed for avoiders in *Figure 2D*. Taken together, these results suggest that some mice, the non-avoiders, rapidly adjust their intake to adapt to changes in nicotine concentration in the bottle. In contrast, for avoiders, aversion to nicotine may involve a learning mechanism that, once triggered, results in prolonged cessation of nicotine consumption.

## Nicotine consumption negatively correlated with the amplitude of nicotine-evoked currents in the IPN

We then investigated the neural correlates of this nicotine aversion. We hypothesized that the IPN, which has been implicated in nicotine aversion and in negative affective states (*Fowler and Kenny, 2014*; *McLaughlin et al., 2017*; *Molas et al., 2017*; *Wills et al., 2022*), might be activated differently by nicotine in avoiders and non-avoiders. Therefore, we used whole-cell patch-clamp recordings in brain slices to assess, at completion of the two-bottle choice task, the functional expression level of nAChRs in IPN neurons. We recorded neurons from the rostral IPN (IPR), because these neurons have high nAChR density (*Hsu et al., 2013*; *Morton et al., 2018*; *Quina et al., 2017*; *Wolfman et al., 2018*). To record nicotine-evoked currents, we used a local puff application of nicotine at a concentration (30 μM) close to the EC50 for heteromeric nAChRs (*Fenster et al., 1997*). We found that the amplitude of nicotine-evoked currents was higher in IPN neurons of avoider mice than in those from non-avoiders (*Figure 2E*). Specifically, there was a negative correlation between the average amplitude of nicotine-evoked current in IPN neurons, and nicotine consumption (measured over the last 24 hr prior to the patch-clamp recording, *Figure 2F*). Because there are many types of neurons in the IPN (*Ables et al., 2017*) with heterogenous responses to nicotine (*Figure 2E*), we verified that there was no anatomical sampling bias between the two groups (*Figure 2—figure supplement 1D*). We also found that there was no relationship between the amplitude of nicotine-induced current and the anatomical localization of the neurons (*Figure 2—figure supplement 1E*). These results indicate that nicotine consumption in mice is negatively linked to the amplitude of the nicotine response in IPN neurons.

## Chronic nicotine treatment alters both nicotinic signaling in the IPN and nicotine consumption

It is still unclear at this stage whether chronic nicotine exposure progressively alters the response of IPN neurons to the drug (non-avoiders being further exposed to high nicotine doses than avoiders) or whether intrinsic differences pre-exist in avoiders and non-avoiders. To determine the effect of chronic nicotine exposure on nAChR current levels in IPN neurons, we passively and continuously exposed mice to nicotine for 4 weeks, using subcutaneously implanted osmotic minipumps. The concentration of nicotine in the minipump (10 mg/kg/day) was chosen to match the average voluntary nicotine

intake in the two-bottle choice task (*Figure 1C*). We then recorded from acute brain slices and found that indeed, prolonged exposure to nicotine reduced the amplitude of nicotine-evoked currents in the IPN of these mice compared to control mice treated with saline (*Figure 3A*; no anatomical sampling bias between the two groups, see *Figure 3—figure supplement 1A*). These results are consistent with the reduced current amplitudes observed in mice that underwent the two-bottle choice task, compared to naive mice in their home cage (*Figure 3—figure supplement 1B*).

Because IPN neurons are mostly silent in brain slices, and in order to preserve the entire circuitry intact, we decided to perform juxtacellular recordings of IPN neurons in vivo, and to characterize their response (expressed in % of variation from baseline) to an intravenous (i.v.) injection of nicotine (30 µg/kg). To the best of our knowledge, no description of in vivo recordings of IPN neurons, and thus no criteria for identification, have been reported as yet, so we solely considered for analysis neurons that were labelled in vivo with neurobiotin and confirmed to be within the IPN. We found that nicotine i.v. injections in naive WT mice induce an increase in the firing rate of IPN neurons compared with saline injection, and that this acute effect of nicotine was reduced after the prolonged (4 weeks) passive exposure of the mice to the drug (*Figure 3B*). The anatomical distribution of recorded neurons was similar between the two groups (*Figure 3—figure supplement 1C*), indicating that the effect of chronic nicotine is unlikely to be due to sampling variability. There was a correlation in both groups between the response to nicotine and the mediolateral (ML), but not the dorso-ventral (DV) coordinates of the neurons (*Figure 3—figure supplement 1D*). Regarding the spontaneous activity of IPN neurons, we found no correlation with their anatomical localization (*Figure 3—figure supplement 1E*), nor any effect of chronic nicotine exposure (*Figure 3—figure supplement 1F*). In both nicotine- and saline-treated animals, the amplitude of the response to nicotine was positively correlated with spontaneous activity: neurons with high basal activity responded to nicotine with a greater change in firing frequency (*Figure 3—figure supplement 1G*). Some IPN neurons responded to nicotine by decreasing their firing rate and, as with nicotine-activated neurons, the response was of smaller amplitude in the chronic nicotine-exposed group (*Figure 3—figure supplement 1H*). Together, these ex vivo and in vivo recordings demonstrate that prolonged exposure to nicotine markedly reduces the amplitude of the drug response in mouse IPN neurons.

To verify the hypothesis that altered cholinergic activity in IPN neurons impacts nicotine aversion, we performed a series of experiments. First, we evaluated the consequence of prolonged nicotine exposure on nicotine consumption. Mice were implanted with an osmotic minipump to passively deliver nicotine and, after 20 days, were subjected to a modified two-bottle task that involved a direct exposure to a high concentration of nicotine (100 µg/ml, *Figure 3C*). We chose this protocol to avoid the confounding effects of a gradual exposure to nicotine, and to evoke strong aversion in the mice. We found that mice pretreated with saline (controls) avoided nicotine (*Figure 3C*), which may indicate aversion to such high nicotine concentration. In contrast, mice pretreated with nicotine did not avoid nicotine, suggesting that they may have developed a tolerance for the aversive effects of the drug. Looking at the data day by day, we observed that control mice abruptly decreased their percent consumption when nicotine was introduced, while for nicotine-treated animals, the decrease was more gradual over the course of the 4 days (*Figure 3—figure supplement 1I*). Overall, nicotine consumption was greater in the nicotine-treated group than in the saline-treated group (*Figure 3C*). Focusing on individuals, we observed that a single saline-pretreated mouse (1/23) increased its percent consumption when nicotine was introduced in the task, while the vast majority of the mice actively avoided nicotine (close to the 0% Nicotine apex). In contrast, a marked proportion of the mice treated with nicotine (8/25) increased their percent consumption when nicotine was introduced (*Figure 3D* p-value = 0.037 Chi-squared). The two groups were identical in the water/water session (*Figure 3E*, top). However, in the water/nicotine session, mice pretreated with nicotine showed a greater distance from the 0% nicotine consumption apex, and a shorter distance from the 100% nicotine consumption apex than mice pretreated with saline (*Figure 3E*, bottom), indicating decreased aversion and increased preference for nicotine in the nicotine-pretreated group. Overall, these electrophysiological and behavioral data demonstrate that prolonged exposure to nicotine both decreases nicotine efficacy in the IPN, and also decreases aversion to nicotine in individuals, leading to increased drug use. However, chronic nicotine may produce adaptations in other brain circuits, and it remains to be demonstrated whether the neurophysiological changes observed in the IPN are causally related to the variations in aversion sensitivity.

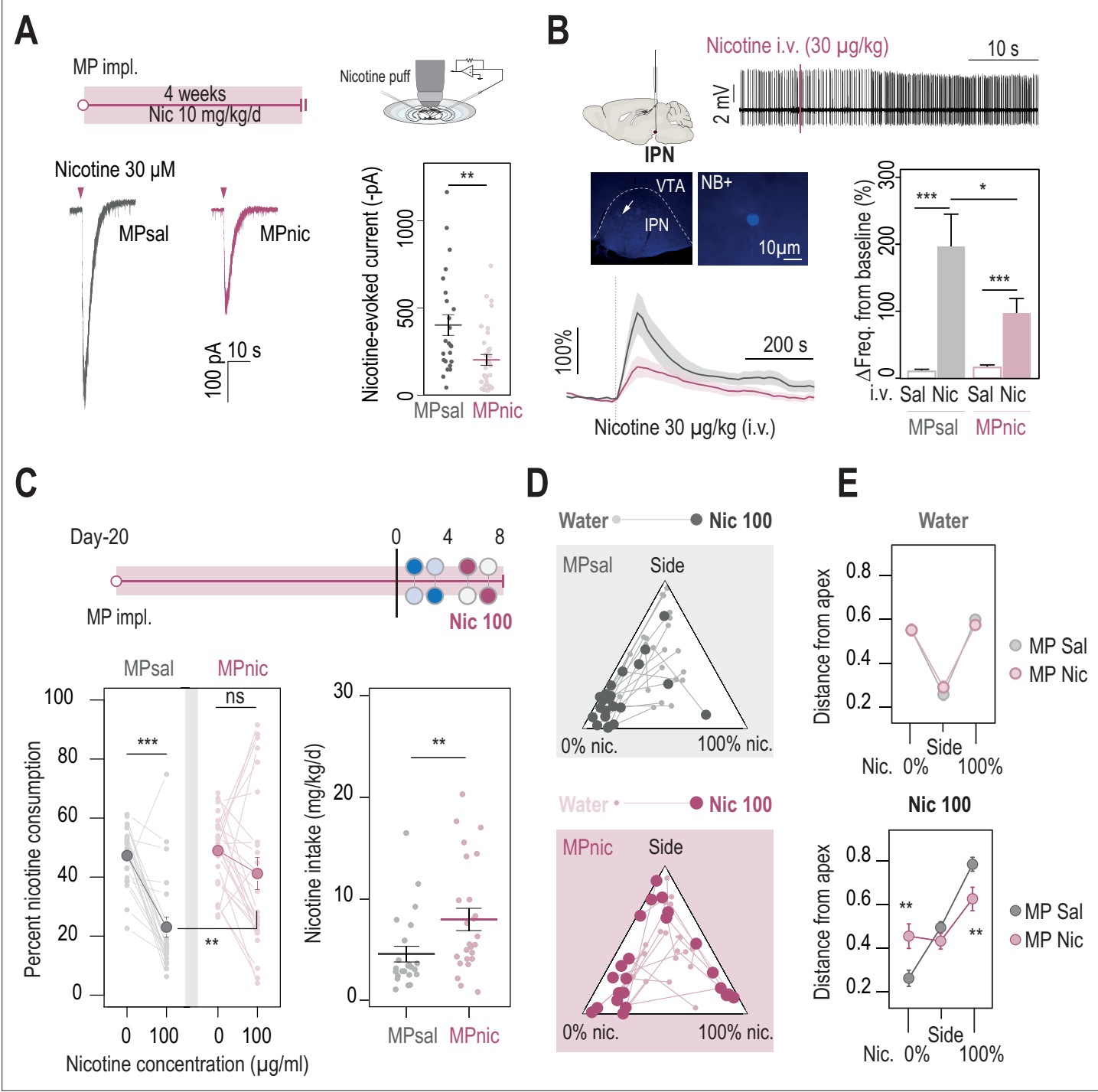

**Figure 3.** Chronic nicotine treatment altered both nAChR expression levels in the IPN and nicotine intake in WT mice. (**A**) Top, passive nicotine treatment protocol. Mice were implanted subcutaneously with an osmotic minipump (MP) that continuously delivers 10 mg/kg/day of nicotine. After 4 weeks of treatment, nicotine-evoked responses in IPN neurons were recorded in whole-cell voltage-clamp mode (–60 mV) from IPN slices. Bottom, representative recordings (left) and average current amplitudes (right) following a puff application of nicotine (30 μM, 200ms) in IPN neurons of mice treated with either saline (n=24 neurons from 3 mice, I = –401±59 pA) or nicotine (n=34 neurons from 4 mice, I = –202±37 pA). Nicotine treatment reduced the amplitude of nicotine-evoked currents in IPN neurons (Mann-Whitney test, p=0.001). (**B**) In vivo juxtacellular recordings of nicotine-evoked responses in IPN neurons of saline- and nicotine-treated animals. Top, representative electrophysiological recording of an IPN neuron, during an i.v. injection of nicotine (30 μg/kg). Middle, post-recording identification of neurobiotin-labeled IPN neurons by immunofluorescence. Bottom, average time course and average amplitude of the change in firing frequency from baseline after an i.v. injection of saline and nicotine (30 μg/kg), for IPN neurons from saline- and nicotine-treated mice. Right, firing rate variation from baseline induced by nicotine or saline injection in IPN neurons from saline- (n=12

*Figure 3 continued on next page*

*Figure 3 continued*

neurons from 6 mice) or nicotine-treated animals (n=20 neurons from 13 mice). Responses were decreased by chronic exposure to nicotine (p=0.035, Mann-Whitney test). All neurons were confirmed to be located within the IPN using juxtacellular labeling with neurobiotin. (**C**) Top, modified two-bottle choice protocol used to evaluate the impact of a long-term exposure to nicotine on drug intake. Mice were implanted subcutaneously with a minipump that delivered 10 mg/kg/day of nicotine continuously, for 20 days before performing the modified two-bottle choice task. After 4 days of water vs. water habituation, mice were directly exposed to a high concentration of nicotine (100 µg/ml). Bottom, percent nicotine consumption and nicotine intake at 0 and 100 µg/ml of nicotine, for mice under a chronic treatment of nicotine or saline. The saline-treated group displayed a decrease in percent nicotine consumption (n=23, from 47.3±2.0%–23.0 ± 3.4%, p=1.7e-05, Mann-Whitney paired test), but not the nicotine-treated group (n=25, from 48.9±2.4 to 41.2 ± 5.5%, p=0.16, Mann-Whitney paired test). Overall, the saline-treated group displayed a lower percent nicotine consumption (p=0.003, Mann Whitney) and lower nicotine intake than the nicotine-treated group (p=0.004, Mann-Whitney). (**D**) Pseudo-ternary diagrams representing each saline- and nicotine-treated mouse for its nicotine consumption index over its side bias index. Small dots correspond to the habituation period (water vs. water) and bigger dots to the condition with 100 µg/ml of nicotine in one bottle. (**E**) Average distance from each apex in the water vs. water (top) and water vs. nicotine 100 µg/ml conditions (bottom). Saline-treated, but not nicotine-treated mice developed a strategy to avoid nicotine (p$_{Sacc}$ = 0.013, p$_{Side}$ = 0.27 p$_{Nic}$=0.013, Mann-Whitney test with Holm-Bonferroni correction). In all figure panels, nicotine-treated animals are displayed in red and saline-treated (control) animals in grey.

The online version of this article includes the following source data and figure supplement(s) for figure 3:

**Source data 1.** Source data for *Figure 3*.

**Figure supplement 1.** Physiological and behavioral adaptations following chronic nicotine treatment.

## Nicotine avoidance involves β4-containing nAChRs

We turned to mutant mice deleted for the gene encoding the β4 nAChR subunit (*Chrnb4$^{-/-}$* mice), because of the strong and restricted expression of this subunit in the MHb-IPN pathway (*Grady et al., 2009*; *Harrington et al., 2016*; *Shih et al., 2014*). The Chrnb4-Cre transgenic mouse (RRID:MMR-RC_036203-UCD, Gene Expression Nervous System Atlas [GENSAT]), which expresses the enzyme Cre-recombinase under the Chrnb4 promoter, revealed that the β4 subunit was distributed mostly in the rostral part of the IPN (IPR) and in its ventral and central parts (IPC), both of which receive cholinergic inputs from the MHb (*Heintz, 2004*). We found that *Chrnb4$^{-/-}$* mice displayed both greater percent nicotine consumption and nicotine intake than WT animals, with minimal concentration-dependent change in percent nicotine consumption (*Figure 4A*). With respect to individuals, we observed both active and passive nicotine-drinking profiles in *Chrnb4$^{-/-}$* mice, as observed in WT mice. Strikingly, however, none of the *Chrnb4$^{-/-}$* mice (0/13) showed aversion-like behavior at high nicotine concentration, which contrasts with the high proportion of avoiders in WT animals (17/35, *Figure 4B*, *Figure 4—figure supplement 1A*, *P*=0.04 Pearson's Chi squared with Yates' continuity correction). Avoidance to quinine was similar in WT and *Chrnb4$^{-/-}$* mice (*Figure 4—figure supplement 1B*), further suggesting that avoidance to nicotine is not linked to different sensitivities to the bitterness of nicotine. WT mice showed a strong nicotine concentration-dependent adaptation in their behavior, while *Chrnb4$^{-/-}$* mice had a more consistent behavior throughout the task (*Figure 4C* and *Figure 4—figure supplement 1C and D*).

To verify that responses to nicotine were affected in IPN neurons of *Chrnb4$^{-/-}$* mice, we first performed whole-cell patch-clamp recordings. We found that the amplitude of nicotine-evoked currents in the IPN was on average three-fold lower in *Chrnb4$^{-/-}$* than in WT mice (*Figure 4D*), confirming that β4-containing (β4*) nAChRs are the major receptor subtype in the rostral IPN. Anatomical sampling was slightly different between the two groups in the mediolateral axis (*Figure 4—figure supplement 2A*). However, this should not affect the interpretation of the results, as there was no relationship between the mediolateral position and the response to nicotine (*Figure 2—figure supplement 1E*). We also found that prolonged nicotine treatment had no significant effect on the amplitude of nAChR currents in these knock-out mice (*Figure 4D*), suggesting that the downregulation observed after chronic nicotine treatment in the IPN of WT mice mainly affects β4*nAChRs (and not other subtypes such as β2*nAChRs for instance).

We then used in vivo juxtacellular recordings, to assess the role of β4*nAChRs in the response to nicotine in the intact brain. Nicotine i.v. injections (30 µg/kg) resulted in an increase in IPN neuron activity (*Figure 4E*), that was larger in WT mice than in *Chrnb4$^{-/-}$* mice, further demonstrating the important role of β4*nAChRs in the response of the IPN to nicotine. Anatomical repartition of the recorded neurons was similar between the two groups in the medio-lateral (ML) but not in the dorso-ventral (DV) axis (*Figure 4—figure supplement 2B*). However, there was no correlation between the

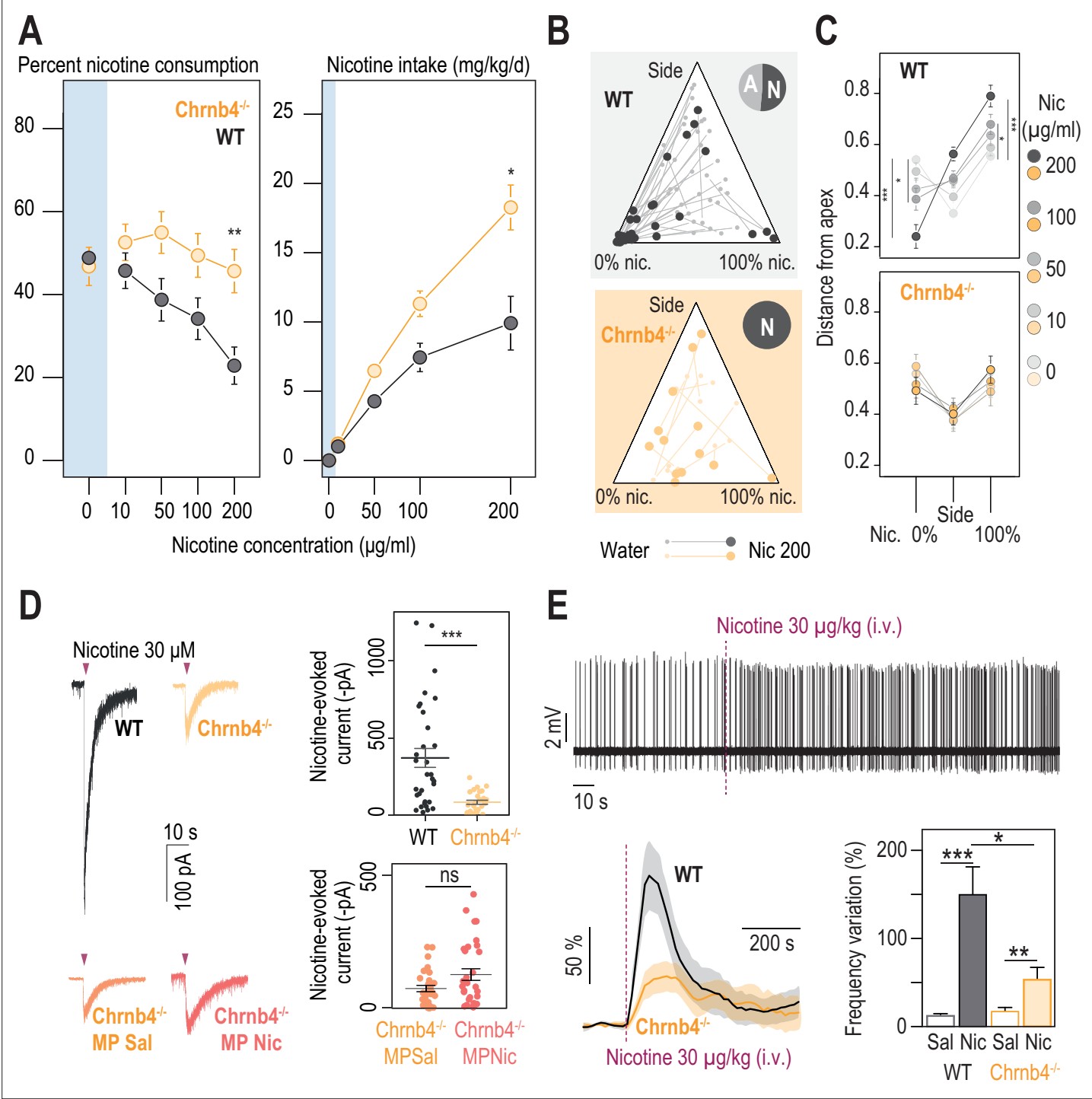

**Figure 4.** β4-containing nicotinic receptors are essential for triggering nicotine aversion in mice. (**A**) Left, average percent nicotine consumption in WT and *Chrnb4*-/- mice for each concentration of nicotine in the two-bottle choice task. WT mice had lower percent nicotine consumption than *Chrnb4*-/- mice (Mann-Whitney test with Holm-Bonferroni correction). WT mice decreased their percent nicotine consumption throughout the task (Friedman test, n=35, df = 4, p<0.001 and Mann-Whitney post-hoc test with Holm-Bonferroni correction) while *Chrnb4*-/- mice displayed a stable percent nicotine consumption (Friedman test, n=13, df = 4, p=0.11). Right, average nicotine intake (mg/kg/day) in *Chrnb4*-/- and WT mice for the different concentrations of nicotine (Friedman test, n=35, df = 3, p<0.001 and Mann-Whitney post-hoc test with Holm-Bonferroni correction). *Chrnb4*-/- mice consumed more nicotine than WT mice (Mann-Whitney test). (**B**) Ternary diagram representing each WT and *Chrnb4*-/- individual for its nicotine consumption index over its side bias index. Small dots correspond to the habituation period (water vs. water) and bigger dots to the condition with 200 µg/ml of nicotine in one bottle. Inserts: pie charts illustrating the proportion of avoiders (A, light grey) and non-avoiders (N, dark grey) for each genotype at the end

*Figure 4 continued on next page*

*Figure 4 continued*

of the task. Note the absence of avoiders in *Chrnb4*[-/-] mice. (**C**) Average distance from each apex at 0, 10, 50, 100, and 200 µg/ml of nicotine (paired Mann-Whitney test with Holm-Bonferroni correction, p(Sacc 0–200)=0.0002, p(Sacc 0–100)=0.03; p(Nic 0–200)=0.0002, p(Sacc 0–100)=0.03). (**D**) Left, representative currents following a puff application of nicotine (30 µM, 200ms) in IPN neurons from naive WT and *Chrnb4*[-/-] mice, or from saline-treated (orange) and nicotine-treated (dark orange) *Chrnb4*[-/-] mice. Right, average nicotine-evoked currents recorded in IPN neurons from naïve WT (n=32 neurons from 5 mice, I = –370±61 pA) and *Chrnb4*[-/-] (n=27 neurons from 4 mice, I = –83±13 pA) mice, and from *Chrnb4*[-/-] chronically treated with either saline (Sal, n=30 neurons from 6 mice, I = –72±11 pA) or nicotine (Nic, n=31 neurons from 5 mice, I = –123±21 pA). *Chrnb4*[-/-] mice presented a large decrease in nicotine-evoked currents (Mann-Whitney test, p=4e-05). Nicotine treatment did not alter nicotine-evoked currents in IPN neurons of *Chrnb4*[-/-] mice (Mann-Whitney test, p=0.15). (**E**) Juxtacellular recordings of nicotine-evoked responses in IPN neurons of naive WT and *Chrnb4*[-/-] mice. Top, representative recording in *Chrnb4*[-/-] mice. Bottom left, average nicotine-evoked responses at 30 µg/kg of nicotine in IPN neurons from WT and *Chrnb4*[-/-] mice. Bottom right, average amplitude of the change in firing frequency from baseline after an i.v. injection of saline and nicotine (30 µg/kg), for IPN neurons from WT (n=18 neurons from 14 mice) and *Chrnb4*[-/-] (n=10 neurons from 8 mice) mice. Nicotine-induced responses were smaller in *Chrnb4*[-/-] than in WT mice (p=0.04, Mann Whitney test). All recorded neurons were neurobiotin-labelled and confirmed to be within the IPN. In all figure panels WT animals are depicted in grey and *Chrnb4*[-/-] mice in yellow. *** p<0.001, ** p<0.01, * p<0.05.

The online version of this article includes the following source data and figure supplement(s) for figure 4:

**Source data 1.** Source data for *Figure 4*.

**Figure supplement 1.** Behavioral differences between WT and *Chrnb4*[-/-] mice.

**Figure supplement 2.** Electrophysiological differences between WT and *Chrnb4*[-/-] mice.

anatomical coordinates of the neurons (whether DV or ML) and their nicotine response (*Figure 4— figure supplement 2C*), allowing us to compare the two groups. Overall, IPN neurons of WT and *Chrnb4*[-/-] mice had similar spontaneous frequencies (*Figure 4—figure supplement 2D and E*), suggesting that β4*nAChRs play little role in regulating the excitability of IPN neurons in anesthetized mice. The amplitude of the nicotine response was positively correlated with the basal firing rate of the neurons for *Chrnb4*[-/-] mice (*Figure 4—figure supplement 2F*), confirming what we observed with WT mice equipped with saline or nicotine minipumps (*Figure 3—figure supplement 1G*). In both WT and *Chrnb4*[-/-] mice, we observed a population of neurons that decreased their firing rate, but with no difference in response amplitude between the two genotypes (*Figure 4—figure supplement 2G*), suggesting that β4*nAChR are mainly involved in increasing, not decreasing, neuronal activity in response to nicotine injection. Collectively, our results in *Chrnb4*[-/-] mice demonstrate the key role of the β4 nAChR subunit in signaling aversion to nicotine, and its predominant function in the activation of the IPN by nicotine.

## β4*nAChRs of the IPN are critically involved in nicotine aversion

β4*nAChRs are enriched in the IPN, yet they are also expressed to some extent in other brain regions. Hence, to directly implicate IPN β4*nAChRs in nicotine aversion, and more generally in nicotine consumption, we targeted re-expression of β4 in the IPN specifically, using lentiviral vectors in *Chrnb4*[-/-] mice (KO-β4[IPN] mice, *Figure 5A*). Mice transduced with eGFP (KO-GFP[IPN] mice) were used as controls. Proper transduction in the IPN was verified using immunohistochemistry after completion of the two-bottle choice task (*Figure 5A*), and mice with expression of GFP in the VTA were excluded from analyses. Transduction of β4, but not of GFP alone, in the IPN increased the amplitude of nicotine-evoked currents (*Figure 5B*) and restored levels found in WT animals (U test, p=0.6, *Figure 5—figure supplement 1A*). Transduction of β4 in the IPN also restored the response to nicotine in vivo (*Figure 5C*). We compared nicotine intake in the two groups of mice in the two-bottle choice task. We found that re-expression of β4 in the IPN of *Chrnb4*[-/-] mice decreased nicotine intake compared to the group of mice transduced with eGFP in the IPN (*Figure 5D*). Overall, nicotine intake was similar in WT and in KO-β4[IPN] animals (p>0.5 for all concentrations), demonstrating the causal role of β4 nAChRs of IPN neurons in nicotine consumption behaviors. At the individual level, the proportion mice that avoided nicotine at 200 µg/kg was very low for KO-GFP[IPN] control mice (2/18), but greater for KO-β4[IPN] mice (8/17, *Figure 5E*, p=0.04 Chi squared). The behavior of KO-GFP[IPN] was steady throughout the task, whereas it was highly nicotine concentration-dependent for KO-β4[IPN] mice (*Figure 5F* and *Figure 5— figure supplement 1B and C*), as already observed with WT mice. Mice with strong percent nicotine consumption were only found in the KO-GFP[IPN] control group. Collectively, these results show that selective re-expression of β4 in the IPN of *Chrnb4*[-/-] mice rescued aversion for nicotine, and highlight

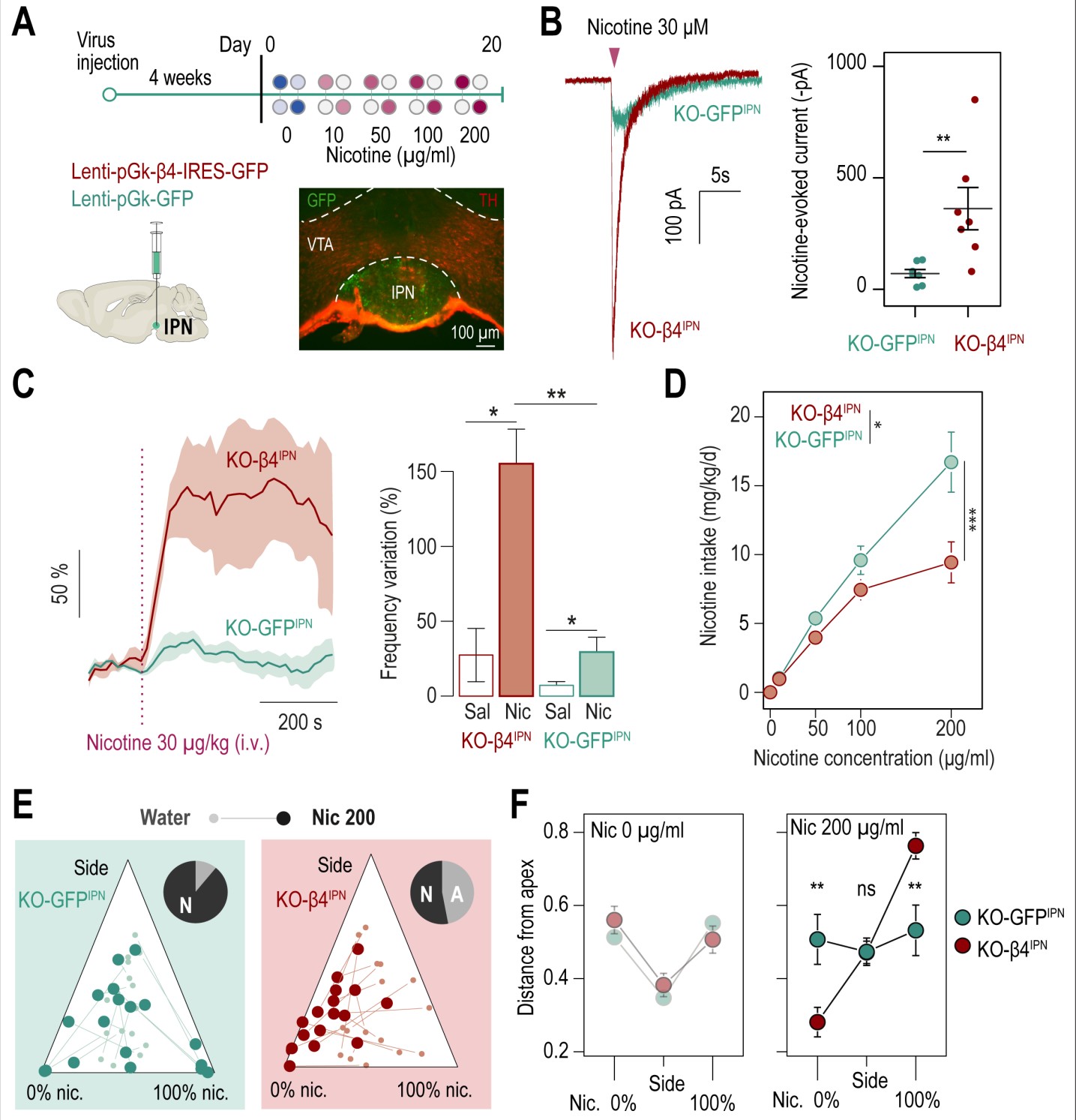

**Figure 5.** β4-containing nAChRs of the IPN are involved in the control of nicotine consumption and in aversion to nicotine in mice. (**A**) Protocol: stereotaxic transduction of the β4 subunit together with GFP (or GFP alone in control mice) in the IPN of *Chrnb4⁻/⁻* mice, and subsequent two-bottle choice task. Bottom right: coronal section highlighting proper viral transduction of lenti-pGK-β4-IRES-GFP in the IPN. (**B**) Validation of the re-expression strategy using whole-cell patch-clamp recordings. Representative currents and average responses following a puff application of nicotine (30 μM, 200ms) on IPN neurons from *Chrnb4⁻/⁻* mice transduced in the IPN with either lenti-pGK-β4-IRES-GFP (KO-β4$^{IPN}$, n=7 neurons from 2 mice, I = –362±95 pA) or lenti-pGK-GFP (KO-GFP$^{IPN}$, n=7 neurons from 1 mouse, I = –71±18 pA; Mann-Whitney test, p=0.004). (**C**) Juxtacellular recordings of nicotine-evoked responses in IPN neurons of naive *Chrnb4⁻/⁻* mice transduced with either GFP (KO-GFP$^{IPN}$) or β4 in the IPN (KO-β4$^{IPN}$). Left, average nicotine-evoked

*Figure 5 continued on next page*

Figure 5 continued

responses at 30 μg/kg of nicotine in IPN neurons from KO-GFP[IPN] and KO-β4[IPN] mice. Right, average amplitude of the change in firing frequency from baseline after an i.v. injection of saline or nicotine (30 μg/kg), for IPN neurons of KO-GFP[IPN] (n=7 neurons from 4 mice) and KO-β4[IPN] mice (n=5 neurons from 3 mice). Nicotine-evoked responses were larger in KO-β4[IPN] than KO-GFP[IPN] mice (p=0.005, Mann Whitney test). (**D**) Average nicotine intake was lower in KO-β4[IPN] than in KO-GFP[IPN] (two-way repeated measure; ANOVA: genotype x dose interaction, F[3, 99]=6.3, ***p<0.001; main effect of dose, F[3, 99]=69.1 ***p<0.001, effect of genotype, F(1, 33)=6.637, *p=0.015). (**E**) Ternary diagram representing each *Chrnb4⁻/⁻* mouse, transduced with either β4 or GFP, and illustrating its nicotine consumption index over its side bias index. Small dots correspond to the habituation period (water vs. water) and bigger dots to the condition with 200 μg/ml of nicotine in one bottle. Inserts: pie charts illustrating the proportion of avoiders (A, light grey) and non-avoiders (N, dark grey) for each condition at the end of the task. (**F**) Average distance from each apex during the two-bottle choice task at 0 and 200 μg/ml of nicotine, for KO-β4[IPN] and KO-GFP[IPN] mice (p$_{Sacc}$ = 0.007, p$_{Nic}$ = 0.008, Mann-Whitney test with Holm-Bonferroni correction). In all figure panels KO-β4[IPN] mice are depicted in red and KO-GFP[IPN] mice (controls) in green. *** p<0.001, ** p<0.01, * p<0.05.

The online version of this article includes the following source data and figure supplement(s) for figure 5:

**Source data 1.** Source data for *Figure 5*.

**Figure supplement 1.** Viral rescue of β4 nAChR subunit expression in the IPN of *Chrnb4⁻/⁻* mice.

the specific role of IPN β4*nAChRs in signaling aversion to nicotine, and in the control of nicotine intake.

## Discussion

We used a two-bottle choice paradigm to assess inter-individual differences in nicotine consumption in mice, and to evaluate how pre-exposure to nicotine modifies drug taking. In this paradigm, mice are given free access to two bottles, only one of which contains nicotine, and allowed to choose which one they drink. Because it does not involve active lever pressing, this method cannot provide as much information as intravenous self-administration about the reinforcing effects of the drug. Yet it remains a simple and classical test to measure the animal's preference (or lack thereof) for the drug, while minimizing stress from handling (*Collins et al., 2012*). In this test, WT mice showed on average no real preference for the nicotine-containing bottle at any of the concentrations tested, as is typically observed (*Matta et al., 2006*). However, it should be noted that, at the individual level, some mice actively tracked the nicotine-containing bottle and showed effective preference. On average, WT mice titrate to a nicotine dose of approximately 10 mg/kg/day, consistent with previous reports (*Antolin-Fontes et al., 2020*; *Fowler et al., 2011*; *Tuesta et al., 2017*), while mutant mice lacking the β4 nAChR subunit did not, resulting in greater nicotine intake, notably at high nicotine concentrations. These results are consistent with the greater intracranial self-administration observed at high nicotine doses in these knock-out mice (*Husson et al., 2020*). Conversely, they are also consistent with the results obtained with transgenic TABAC mice overexpressing the β4 subunit at endogenous sites, which avoid nicotine and consequently consume very little (*Frahm et al., 2011*). Nevertheless, it should be noted that conflicting results have also been reported regarding the role of β4 nAChRs in nicotine consumption. Notably, another study reported that *Chrnb4⁻/⁻* mice exhibit lower intravenous self-administration of nicotine, even though their VTA is more sensitive to nicotine (*Harrington et al., 2016*). Conversely, self-administration is higher in TABAC mice despite reduced activation of the VTA by nicotine (*Gallego et al., 2012*). The increased consumption at high nicotine concentration reported here for *Chrnb4⁻/⁻* mice resembles what was observed in mutant mice either lacking the α5 subunit (*Fowler et al., 2011*) or with low levels of the α3 subunit (*Elayouby et al., 2021*), likely because the α3, α5, and β4 nAChR subunits, which are encoded by the same gene cluster, co-assemble in brain structures, notably the MHb-IPN pathway, to produce functional heteromeric nAChRs.

It is increasingly acknowledged that in mice, as in humans, there is a substantial variability in the susceptibility for developing drug use disorders (*Deroche-Gamonet et al., 2004*; *Dongelmans et al., 2021*; *Garcia-Rivas et al., 2017*; *Juarez et al., 2017*; *Nesil et al., 2011*; *Piazza et al., 1989*; *Siciliano et al., 2019*). Yet, it is still unclear why some individuals are more susceptible than others to become regular users. We discovered important inter-individual differences in nicotine vulnerability: about half of the WT mice, the avoiders, durably quit nicotine at a certain concentration, whereas the other half, the non-avoiders, continued consumption even at high concentration of nicotine, classically described as aversive (*Fowler et al., 2011*). Avoiders displayed variable concentration thresholds required for triggering aversion, and some even developed aversion at the beginning of the two-bottle choice

task, when nicotine concentrations were still low. The nicotine concentration threshold required to trigger aversion differed among avoiders, and was likely not reached in non-avoiders and *Chrnb4⁻/⁻* mice. We conducted experiments with adult male mice only, so it remains an open question whether nicotine aversion varies with sex and age. Importantly, very few mice showed what might be considered titration (plateau consumption), emphasizing the needs to consider individual, as opposed to group, behavior in addiction research.

We also found that the functional expression level of β4-containing nAChRs in the IPN underlies these different sensitivities to the aversive properties of nicotine. Indeed, nicotine aversion was nearly eliminated in *Chrnb4⁻/⁻* mice, none of which quit drinking nicotine, and was restored after selective re-expression of the β4 subunit in the IPN. This local re-expression experiment, together with the absence of correlation between nicotine and quinine aversion, discards the possibility that different sensitivities to the bitter taste of nicotine solutions explain the different drinking profiles of avoiders and non-avoiders. We observed a negative correlation between nicotine consumption and the response of the IPN to the drug: mice consuming large amounts of nicotine displayed lower nicotine-evoked currents in the IPN than mice consuming small amounts of nicotine. Accordingly, *Chrnb4⁻/⁻* mice and nicotine-treated WT mice, both of which display reduced responses to nicotine, consume greater amounts of the drug. It should be noted that there is a variety of neuronal types in the IPN (*Ables et al., 2017*; *García-Guillén et al., 2021*) and our electrophysiological recordings did not address this genetic diversity. That said, we sampled similar populations of neurons across groups, and found little evidence of a relationship between anatomical localization and response to nicotine, allowing for comparisons across conditions. We also found that nAChRs that contain the β4 subunit represent the vast majority of nicotinic receptors in the IPN. Taken together, these results indicate that the expression level of β4-containing nAChRs in the IPN may determine the level of aversion to the drug, and consequently its intake.

We suggest that β4-containing nAChRs, by engaging the IPN circuitry, initiate a primary response to nicotine that, if above a certain threshold, will trigger acute aversion to the drug, impacting the balance between drug reward and aversion to limit drug consumption. Consistent with this, pharmacological or optogenetic stimulation of the MHb-IPN pathway directly produces aversion (*Morton et al., 2018*; *Tuesta et al., 2017*; *Wolfman et al., 2018*), while pharmacological inactivation of this pathway increases nicotine intake (*Fowler et al., 2011*). We further found that aversion to nicotine is not just an acute adaptation to the dose, but can last for days. This is consistent with findings in humans, where an initial unpleasant reaction to cigarettes is associated with a reduced likelihood of continued smoking (*DiFranza et al., 2004*). Such a sustained aversive reaction to nicotine was conditioned, in mice, by nicotine itself, and required β4-containing nAChRs of the IPN for its onset, but most likely involves other brain circuits for its long-term persistence. Identifying the molecular and cellular mechanism of long-term aversion to nicotine in mice will be instrumental to progress in our understanding of human dependence to tobacco.

We did not observe major differences in nicotine intake between avoiders and non-avoiders at the beginning of the two-bottle choice experiment, for low concentrations of nicotine (<100 µg/ml), but we cannot completely rule out pre-existing inter-individual differences that could explain the opposite trajectories taken by the two groups. Indeed, the mice used in this study were isogenic, yet epigenetic changes during development or differences in social status, which are known to affect brain circuits and individual traits (*Torquet et al., 2018*), may affect the responses of the IPN to nicotine. It is indeed tempting to speculate that external factors (e.g. stress, social interactions…) that would affect either the expression level of β4-containing nAChRs in the IPN or the availability of nicotine in the brain (through different metabolic activities for instance), will have a strong impact on nicotine consumption. In addition to pre-existing differences, history of nicotine use could produce long-lasting molecular and cellular adaptations in the IPN circuitry and consequently alter nicotine aversion and consumption. Indeed, we discovered that prolonged nicotine exposure downregulated and/or desensitized β4-containing nAChRs of the IPN, as evidenced by the decreased response to nicotine both ex vivo and in vivo, and the absence of effect in Chrnb4⁻/⁻ mice.

This observed functional downregulation of nAChR currents contrasts with previous findings. Chronic nicotine was shown to cause functional upregulation of nicotinic currents in MHb neurons of mice (*Arvin et al., 2019*; *Banala et al., 2018*; *Pang et al., 2016*; *Shih et al., 2015*; *Jin et al., 2020*) and rats (*Jin et al., 2020*), as well as in IPN neurons of mice (*Zhao-Shea et al., 2013*) and rats (*Tapia*

*et al., 2022*). However, it should be noted that upregulation is highly cell-type as well as receptor-subtype dependent (*Nashmi et al., 2007*; *Shih et al., 2015*; *Zhao-Shea et al., 2013*). For instance, in the mouse IPN, functional upregulation was shown only in SST-positive neurons, which constitute a small fraction of IPN neurons, and no increase in β4 subunit expression was reported in SST-negative neurons (*Shih et al., 2015*). Moreover, in cell culture, it was found that β2-, but not β4-containing nAChRs were upregulated after nicotine treatment (*Wang et al., 1998*), which agrees with our findings. Upregulation is likely due to an increase in receptor number (*Banala et al., 2018*), yet it can be masked by receptor desensitization, which occurs during prolonged nicotine treatment. In some of the reports of nicotine-induced upregulation, currents were recorded only after nicotine withdrawal (*Pang et al., 2016*), leaving enough time for the receptors to recover from desensitization. Here, we recorded responses to nicotine in vivo, while nicotine was still present, and found that they were reduced in amplitude, which fits with the downregulation we observed in slices. Furthermore, our behavioral data in nicotine-treated WT and *Chrnb4$^{-/-}$* mice are in complete agreement: both displayed reduced responses to nicotine (ex vivo and in vivo) in IPN neurons, as well as increased nicotine intake compared to naive, WT animals.

We suggest that long-term exposure to nicotine decreases the likelihood to reach the threshold at which mice develop aversion to the drug, which ultimately leads to increased drug consumption. In other words, nicotine intake history weakens the ability of nicotine to induce aversion in mice. In most nicotine replacement therapies, such as gums or patches, nicotine is slowly administered over prolonged periods of time, ostensibly to mitigate negative emotional reactions elicited by nicotine withdrawal (*Hartmann-Boyce et al., 2018*). However, our data indicate that mice under prolonged nicotine administration will also develop tolerance to the aversive effects of nicotine, underscoring the necessity to develop alternative medical approaches.

## Materials and methods

### Animals

Eight- to sixteen-week-old wild-type C57BL/6 J (Janvier labs, France) and *Chrnb4* knock-out (*Chrnb4$^{-/-}$*) male mice (Pasteur Institute, Paris) (*Xu et al., 1999*) were used for this study. *Chrnb4$^{-/-}$* mice were backcrossed onto C57BL/6 J background for more than twenty generations. Mice were maintained on a 12 hr light-dark cycle. All experiments were performed in accordance with the recommendations for animal experiments issued by the European Commission directives 219/1990, 220/1990 and 2010/63, and approved by Sorbonne Université.

### Two-bottle choice experiment

Mice single-housed in a home cage were presented with two bottles of water (Volvic) for a habituation period of 4 days. After habituation, mice were presented with one bottle of saccharine solution (2%, Sigma Aldrich) and one bottle of nicotine (free base, Sigma Aldrich) plus saccharine (2%) solution diluted in water (adjusted to pH ~7.2 with NaOH). Unless otherwise noted, four different concentrations of nicotine were tested consecutively (10, 50, 100, and 200 μg/ml) with changes in concentration occurring every 4 days. For the two-bottle aversion task, a single nicotine concentration (100 μg/ml) was used after the habituation period. Bottles were swapped every other day to control for side preference. The drinking volume was measured every minute with an automated acquisition system (TSE system, Germany). Nicotine intake was calculated in mg of nicotine per kilogram of mouse body weight per day (mg/kg/day). To minimize stress from handling, mice were weighed every other day, since we found their weigh to be sufficiently stable over 2 days. Percent nicotine consumption was calculated as the volume of nicotine solution consumed as a percentage of the total fluid consumed. Mice showing a strong side bias (preference <20% or>80%) in the habituation period were not taken into account for the analyses.

For the pseudo-ternary plot analyses, we determined the percent nicotine consumption on the left-hand side (%c1) and the percent nicotine consumption on the right-hand side (%c2), for each nicotine concentration and each animal. We then calculated the nicotine consumption and side bias indexes, by plotting the minimum min(%c1, %c2) against the maximum max(%c1, %c2), and used a 90° rotation to obtain the pseudo-ternary plot. In this plot, the three apices represent mice that avoid nicotine

on both sides (0% nic.), mice that track nicotine on both sides (100% nic.), and finally mice that drink solely one side (side biased).

## Prolonged treatment with nicotine
Osmotic minipumps (2004, Alzet minipump) were implanted subcutaneously in 8-week-old mice anesthetized with isoflurane (1%). Minipumps continuously delivered nicotine (10 mg/kg/day) or saline (control) solution with a rate of 0.25 µl/hr during 4 weeks.

## Brain slice preparation
Mice were weighed and then anaesthetized with an intraperitoneal injection of a mixture of ketamine (150 mg/kg, Imalgene 1000, Merial, Lyon, France) and xylazine (60 mg/kg, Rompun 2%, Bayer France, Lyon, France). Blood was then fluidized by an injection of an anticoagulant (0.1 mL, heparin 1000 U/mL, Sigma) into the left ventricle, and an intra-cardiac perfusion of ice-cold (0–4°C), oxygenated (95% $O_2$/5% $CO_2$) sucrose-based artificial cerebrospinal fluid (SB-aCSF) was performed. The SB-aCSF solution contained (in mM): 125 NaCl, 2.5 KCl, 1.25 $NaH_2PO_4$, 5.9 $MgCl_2$, 26 $NaHCO_3$, 25 sucrose, 2.5 glucose, 1 kynurenate (pH 7.2). After rapid brain sampling, slices (250 µm thick) were cut in SB-aCSF at 0–4°C using a Compresstome slicer (VF-200, Precisionary Instruments Inc). Slices were then transferred to the same solution at 35 °C for 10 min, then moved and stored in an oxygenated aCSF solution at room temperature. The aCSF solution contained in mM: 125 NaCl, 2.5 KCl, 1.25 $NaH_2PO_4$, 2 $CaCl_2$, 1 $MgCl_2$, 26 $NaHCO_3$, 15 sucrose, 10 glucose (pH 7.2). After minimum 1 hr of rest, slices were placed individually in a recording chamber at room temperature and infused continuously with aCSF recording solution at a constant flow rate of about 2 ml/min.

## Ex vivo patch-clamp recordings of IPN neurons
Patch pipettes (5–8 MΩ) were stretched from borosilicate glass capillaries (G150TF-3, Warner instruments) using a pipette puller (Sutter Instruments, P-87, Novato, CA) and filled with a few microliters of an intracellular solution adjusted to pH 7.2, containing (in mM): 116 K-gluconate, 20 HEPES, 0.5 EGTA, 6 KCl, 2 NaCl, 4 ATP, 0.3 GTP and 2 mg/mL biocytin. Biocytin was used to label the recorded neurons. The slice of interest was placed in the recording chamber and viewed using a white light source and a upright microscope coupled to a Dodt contrast lens (Scientifica, Uckfield, UK). Neurons were recorded from the dorsal (IPDL) and rostral (IPR) parts on the IPN. Whole-cell configuration recordings of IPN neurons were performed using an amplifier (Axoclamp 200B, Molecular Devices, Sunnyvale, CA) connected to a digitizer (Digidata 1550 LowNoise acquisition system, Molecular Devices, Sunnyvale, CA). Signal acquisition was performed at 10 kHz, filtered with a lowpass (Bessel, 2 kHz) and collected by the acquisition software pClamp 10.5 (Molecular Devices, Sunnyvale, CA). Nicotine tartrate (30 µM in aCSF) was locally and briefly applied (200ms puffs) using a puff pipette (glass pipette ~3 µm diameter at the tip) positioned about 20–30 µm from the soma of the neuron. The pipette was connected to a Picospritzer (PV-800 PicoPump, World Precision Instruments) controlled with pClamp to generate transient pressure in the pipette (~2 psi). Nicotine-evoked currents were recorded in voltage-clamp mode at a membrane potential of –60 mV. All electrophysiology traces were extracted and preprocessed using Clampfit (Molecular Devices, Sunnyvale, CA) and analyzed with R.

## In vivo electrophysiology
Mice were deeply anesthetized with chloral hydrate (8%, 400 mg/kg) and anesthesia was maintained throughout the experiment with supplements. Catheters were positioned in the saphenous veins of the mice to perform saline or nicotine intravenous injections. Nicotine hydrogen tartrate salt (Sigma-Aldrich) was dissolved in 0.9% NaCl solution and pH was adjusted to 7.4. The nicotine solution was injected at a dose of 7.5, 15, and 30 µg/kg. Borosilicate glass capillaries (1.5 mm O.D. / 1.17 mm I.D., Harvard Apparatus) were pulled using a vertical puller (Narishige). Glass pipettes were broken under a microscope to obtain a ~1 µm diameter at the tip. Electrodes were filled with a 0.5% NaCl solution containing 1.5% of neurobiotin tracer (AbCys) yielding impedances of 6–9 MΩ. Electrical signals were amplified by a high-impedance amplifier (Axon Instruments) and supervised through an audio monitor (A.M. Systems Inc). The signal was digitized, sampled at 25 kHz and recorded on a computer using Spike2 (Cambridge Electronic Design) for later analysis. IPN neurons were recorded in an area corresponding to the following stereotaxic coordinates (4–5° angle): 3.3–3.6 mm posterior to

bregma, 0.2–0.45 mm from medial to lateral and 4.3–5 mm below the brain surface. A 5 min-baseline was recorded prior to saline or nicotine i.v. injection. For the dose-response experiments, successive randomized injections of nicotine (or saline) were performed, interspaced with sufficient amount of time (>10 min) to allow the neuron to return to its baseline.

## Stereotaxic viral injections

8-week-old mice were injected in the IPN with a lentivirus that co-expresses the WT β4 subunit together with eGFP (or only eGFP for control experiments) under the control of the PGK promoter. Lentiviruses were produced as previously described (*Maskos et al., 2005*). For viral transduction, mice were anaesthetized with a gas mixture containing 1–3% isoflurane (IsoVet, Pyramal Healthcare Ltd., Nothumberland, UK) and placed in a stereotactic apparatus (David Kopf Instruments, Tujunga, CA). Unilateral injections (0.1 µl/min) of 1 µl of a viral solution (Lenti.pGK.β4.IRES.eGFP, titer 150 ng/µl of p24 protein; or Lenti.pGK.eGFP, titer 75 ng/µl of p24 protein) were performed using a cannula (diameter 36 G, Phymep, Paris, France). The cannula was connected to a 10 µL Hamilton syringe (Model 1701, Hamilton Robotics, Bonaduz, Switzerland) placed in a syringe pump (QSI, Stoelting Co, Chicago, IL, USA). Injections were performed in the IPN at the following coordinates (5° angle): from bregma ML - 0.4 mm, AP - 3.5 mm, and DV: - 4.7 mm (according to Paxinos & Franklin). Electrophysiological recordings were made at least 4 weeks after viral injection, the time required for the expression of the transgene, and proper expression was subsequently checked using immunohistochemistry.

## Immunocytochemical identification

Immunostaining was performed as described in *Durand-de Cuttoli et al., 2018, with the following* primary antibodies: anti-tyrosine hydroxylase 1:500 (anti-TH, Sigma, T1299) and chicken anti-eGFP 1:500 (Aveslab, AB_10000240). Briefly, serial 60 µm-thick sections of the midbrain were cut with a vibratome. Slices were permeabilized for one hour in a solution of phosphate-buffered saline (PBS) containing 3% bovine serum albumin (BSA, Sigma; A4503). Sections were incubated with primary antibodies in a solution of 1.5% BSA and 0.2% Triton X-100 overnight at 4 °C, washed with PBS and then incubated with the secondary antibodies for 1 hr. The secondary antibodies were Cy3-conjugated anti-mouse (1:500 dilution) and alexa488-conjugated anti-chicken (1:1000 dilution; Jackson Immu-noResearch, 715-165-150 and 703-545-155, respectively). For the juxtacellular immunostaining, the recorded neurons were identified with the addition of AMCA-conjugated streptavidin (1:200 dilution) in the solution (Jackson ImmunoResearch). Slices were mounted using Prolong Gold Antifade Reagent (Invitrogen, P36930). Microscopy was carried out either with a confocal microscope (Leica) or with an epifluorescence microscope (Leica), and images were captured using a camera and analyzed with ImageJ.

## Statistical analysis

All statistical analyses were computed using R (The R Project, version 4.0.0). Results were plotted as a mean ± s.e.m. The total number (n) of observations in each group and the statistics used are indicated in figure legends. Classical comparisons between means were performed using parametric tests (Student's T-test, or ANOVA for comparing more than two groups when parameters followed a normal distribution [Shapiro test p>0.05]), and non-parametric tests (here, Mann-Whitney or Friedman) when the distribution was skewed. Multiple comparisons were corrected using a sequentially rejective multiple test procedure (Holm-Bonferroni correction). All statistical tests were two-sided. p>0.05 was considered not to be statistically significant.

## Acknowledgements

The authors thank Ines Centeno-Lemaire (Sorbonne Université, Paris, France) for her help with behavioral tests, Jean-Pierre Hardelin (ESPCI, Paris, France) for critical reading of the manuscript, and the animal facilities at Institut de Biology Paris Seine (IBPS, Paris, France) and ESPCI Paris. Agence Nationale de la Recherche (ANR-21-CE16-0012 CHOLHAB to AM, and ANR-17-CE16-0016 SNP-NIC to PF) Fondation pour la Recherche Médicale (Equipe FRM EQU201903007961 to PF and a PhD Fellowship to JJ) Institut National du Cancer Grant TABAC-16-022, TABAC-19-02 and SPA-21–002 (to PF). Memolife labex starting package (to PF and AM). Fundamental research prize from the Fondation Médisite for neuroscience (AM). Fourth-year PhD fellowship from Fondation pour la Recherche

Médicale (FDT201904008060 to SM and FDT20170437427 to RDC). Fourth-year PhD fellowship from the Biopsy Labex (CN). Fourth-year PhD fellowship from the Memolife Labex (JJ).

## Additional information

### Funding

| Funder | Grant reference number | Author |
| --- | --- | --- |
| Agence Nationale de la Recherche | ANR-21-CE16-0012 CHOLHAB | Alexandre Mourot |
| Agence Nationale de la Recherche | ANR-17-CE16-0016 SNP-NIC | Philippe Faure |
| Fondation pour la Recherche Médicale | FRM EQU201903007961 | Philippe Faure |
| Institut National Du Cancer | TABAC-16-022 | Philippe Faure |
| Institut National Du Cancer | TABAC-19-02 | Philippe Faure |
| Institut National Du Cancer | SPA-21-002 | Philippe Faure |
| Fondation de France | Prix Médisite | Alexandre Mourot |
| Fondation pour la Recherche Médicale | FDT201904008060 | Sarah Mondoloni |
| Fondation pour la Recherche Médicale | FDT20170437427 | Romain Durand-de Cuttoli |
| Labex Biopsy | | Claire Nguyen |
| Labex Memolife | | Joachim Jehl Alexandre Mourot |
| Fondation pour la Recherche Médicale | | Joachim Jehl |

The funders had no role in study design, data collection and interpretation, or the decision to submit the work for publication.

### Author contributions

Sarah Mondoloni, Conceptualization, Software, Formal analysis, Validation, Investigation, Visualization, Methodology, Writing – review and editing; Claire Nguyen, Maria Ciscato, Fabio Marti, Formal analysis, Investigation; Eléonore Vicq, Joachim Jehl, Romain Durand-de Cuttoli, Stefania Tolu, Investigation; Nicolas Torquet, Software; Stéphanie Pons, Uwe Maskos, Resources; Philippe Faure, Conceptualization, Supervision, Funding acquisition, Validation, Methodology, Writing – review and editing; Alexandre Mourot, Conceptualization, Supervision, Funding acquisition, Validation, Visualization, Methodology, Writing - original draft, Writing – review and editing

### Author ORCIDs

Sarah Mondoloni  http://orcid.org/0000-0002-6134-3715
Claire Nguyen  http://orcid.org/0000-0002-0347-3626
Joachim Jehl  http://orcid.org/0000-0001-9821-7619
Nicolas Torquet  http://orcid.org/0000-0001-9032-193X
Philippe Faure  http://orcid.org/0000-0003-3573-4971
Alexandre Mourot  http://orcid.org/0000-0002-8839-7481

### Ethics

All experiments were performed in accordance with the recommendations for animal experiments issued by the European Commission directives 219/1990, 220/1990 and 2010/63, and approved by Sorbonne Université.

### Decision letter and Author response

Decision letter https://doi.org/10.7554/eLife.80767.sa1

Author response https://doi.org/10.7554/eLife.80767.sa2

## Additional files

### Supplementary files
• MDAR checklist

### Data availability
All data generated or analysed during this study are included in the manuscript and supporting files. Source data have been provided for all the figures.

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
