## [Editor Report]

The vulnerability to adaptations in nicotine addiction is largely determined by individual differences in neural activity controlling nicotine aversion. In the current study, Mondoloni and colleagues differentiated individual mice into "avoiders" and "non-avoiders" based on their nicotine drinking behavior in two-bottle choice tests, and identified a nicotinic receptor, β4 nAChR, in the interpeduncular nuclei as a key substrate for mediating nicotine aversion. This finding has important implications for understanding individual differences in drug addiction.

---

## [Decision Letter]

**Decision letter after peer review:**

[Editors’ note: the authors submitted for reconsideration following the decision after peer review. What follows is the decision letter after the first round of review.]

Thank you for submitting the paper "Prolonged nicotine exposure reduces aversion to the drug in mice by altering nicotine transmission in the antiparticle nucleus" for consideration by *eLife*. Your article has been reviewed by 3 peer reviewers, one of whom is a member of our Board of Reviewing Editors, and the evaluation has been overseen by a Senior Editor. The reviewers have opted to remain anonymous.

Comments to the Authors:

We are sorry to say that, after consultation with the reviewers, we have decided that this work will not be considered further for publication by *eLife*.

The three reviewers raised many concerns regarding the experimental designs, data analysis, controls, statistics, sex bias in experiments, insufficiency in electrophysiological properties, and gene-phenotype relationship. Specifically, their main concerns included that the oral nicotine drinking model does not test nicotine reinforcement, and the electrophysiological recordings lack cell specificity, etc. We hope that you will find our reviewers' comments helpful and that you will soon receive a more encouraging response elsewhere.

*Reviewer #1 (Recommendations for the authors):*

1. In overall, the concern of the reviewer is the insufficiency in addressing the differences in beta4-nAchR of "avoiders" versus "non-avoiders". How is Î²4 distributed in IPN? What is the cell type specificity? The "avoiders'" and "non-avoiders'" behavioral and electrophysiological relationship with Î²4 expression should be addressed.

2. The authors used a pseudo-ternary plot to differentiate nicotine "avoiders" and "non-avoiders". The main behavioral and electrophysiological factors that differentiate "avoiders" from "non-avoiders" can be but were not fully drawn, from the data presented. Also, there is still apparent heterogeneity between "avoiders" and "non-avoiders" (Figure 2A). How these differences relate to IPN nicotine-evoked currents, and if the power of the plot is sufficient for dissecting the population, can be further explored.

4. The change of in vivo activity of the IPN in Î²4 KO or OE rescue should be documented.

5. The individual difference in sensitivity to the bitterness of nicotine should be excluded by using nicotine/quinine vs. quinine preference tests.

*Reviewer #2 (Recommendations for the authors):*

– The authors need to provide the sex of the mice used in each experiment. If both sexes were used across the study, the authors should describe how and if the estrus cycle was monitored.

– The authors should describe the power analysis used to determine the sample sizes of the current data set. For example, it is unclear why a double number of mice/neurons were recorded in N-Av while the related s.e.m. is 50% lower (Figure 2) compared to the Av group.

– The IPn is a heterogeneous brain structure. Also, the authors should further describe the location of the recorded neurons and the viral expression.

– The Y-axis in supplement figure 1D doesn't align well.

*Reviewer #3 (Recommendations for the authors):*

– The oral model is a major weakness. I suggest using an alternative model (and/or species) with more translational relevance. Relatedly, it will be important for this work to use concentrations and doses of nicotine that are relevant to human tobacco users.

– The fact that there are a variety of identifiable types of neurons in the IPN is a very important issue because this work did not take this into account. Differences in nicotine receptor function, noted by the authors, could be simply from sampling variability rather than due to the action of chronic nicotine. A version of this paper that makes a strong impact on the field will need to link changes in nAChR currents to specific types of neurons in the IPN.

– This paper had aspects that were not novel, which prevented the paper from making a strong impact on the field. The manuscript needs a limitations section that provides a measured assessment of the weaknesses of the paper.

[Editors’ note: further revisions were suggested prior to acceptance, as described below.]

Thank you for resubmitting your work entitled "Prolonged nicotine exposure reduces aversion to the drug in mice by altering nicotinic transmission in the interpeduncular nucleus" for further consideration by *eLife*. Your revised article has been evaluated by Kate Wassum (Senior Editor) and a Reviewing Editor.

The manuscript has been improved but there are some minor points that need to be addressed. Reviewer #3 mentioned that beta4 functional responses tend to be up-regulated following contingent or non-contingent nicotine treatment in the MHb/IPN pathway in multiple species. Elaboration in the discussion of this point is recommended.

*Reviewer #3 (Recommendations for the authors):*

The present revision is improved over the previous version and has addressed most of the review concerns. A key finding remains at odds with other findings in the literature, though. The manuscript reports that chronic, passive exposure to nicotine results in down-regulation of nAChRs, presumably beta4-containing receptors (identification of beta4-containing was inferred, not demonstrated with pharmacology). The paper discusses that their results are contradictory to those of Arvin et al. and offer speculation that this is due to methodological differences. The finding of nAChR down-regulation following nicotine treatment is actually contradictory to findings reported in many papers, across several labs, and across several species/preparations. Consider the following regarding the medial habenula / IPN pathway (where beta4 receptors are enriched) and in cultured cells:

1. Shih et al. (Mol Pharm 2015). Chronic nicotine treatment in mice (mini pumps) results in upregulation of nAChRs in medial habenula, a brain area with similar beta4-containing nAChRs.

2. Banala et al. (Nature Methods 2018). MHb nAChR upregulation following chronic nicotine in mice (nicotine drinking water).

3. Jin et al. (eNeuro, 2020). Upregulation of nAChRs in MHb following nicotine self-administration in rats, shown with ACh puff experiments.

4. Tapia et al. (Neuropharm. 2022). Nicotine self-administration in rats results in upregulation of nAChRs in IPN – demonstrated with ACh puff experiments, not uncaging.

5. Zhao-Shea et al. (Current Biology 2013). Beta4-containing receptors are upregulated (functionally and mRNA) in dorsal IPN in response to chronic nicotine treatment in mice (nicotine drinking water).

6. Mazzo et al. (J. Neurosci. 2013). Alpha3beta4 receptors are upregulated by nicotine treatment in vitro.

Thus, the finding of downregulation of nAChRs in the present manuscript is interesting. The paper would be strengthened if there was an improved discussion of this issue.

---

## [Author Response]

[Editors’ note: The authors appealed the original decision. What follows is the authors’ response to the first round of review.]

Reviewer #1 (Recommendations for the authors):1. In overall, the concern of the reviewer is the insufficiency in addressing the differences in beta4-nAchR of "avoiders" versus "non-avoiders". How is Î²4 distributed in IPN? What is the cell type specificity? The "avoiders'" and "non-avoiders'" behavioral and electrophysiological relationship with Î²4 expression should be addressed.

Our work demonstrates for the first time the role of IPN b4 nAChRs in the sensitivity to the aversive properties of nicotine. Unfortunately, there is no good antibody specific to b4. However, images from a transgenic mouse line (Chrnb4-Cre, or OL57) that expresses the enzyme Cre recombinase under the promoter of the b4 nAChR gene are available on the following Gensat webpage (see Author response image 1, the mouse line mouse line is however no longer alive in the Gensat colony):

http://www.gensat.org/creGeneView.jsp?founder_id=84314&gene_id=400&backcrossed=false

These images show that b4 is expressed in IPN neurons. More specifically, labeled cell bodies are found in the rostral part of the IPN (IPR), and in its ventral and central parts (IPC), both of which receive cholinergic inputs from the medial habenula (MHb). In contrast, no b4-labelled cells are not found in the lateral parts of the IPN (IPL and IPDL), two subnuclei that do not receive cholinergic inputs from the MHb (Frahm et al. *eLife* 2015). We could potentially, if allowed, add these images to supplementary figures (or eventually a link to these images in the text), should the reviewers and editors deem it necessary.

**Author response image 1. sa2fig1:** 

In our experiments, we have recorded either solely from the IPR (whole-cell patch clamp experiments) or mostly from the IPR and IPC (juxtacellular electrophysiology). We now provide maps of the recorded neurons for all figures. Several of our findings confirm that b4 constitutes the large majority of functional nAChRs in IPN neurons. First, we found, using patch clamp recordings, that nicotine-evoked currents are reduced by about 80% in b4^-/-^ mice (Figure 4D). Second, the in vivo response to nicotine is also largely reduced (by about 65%) in mice deleted for b4 (Figure 4E). Finally, b4^-/-^ mice have a strong phenotype (they consume large amounts of nicotine and show no avoidance to the drug) and local re-expression of b4 in the IPN restores WT consumption profiles.Furthermore, we found that the amplitude of the response to nicotine (whether in patch or in vivo) were mostly not correlated with the anatomical localizations of the neurons (see Figure 2—figure supplement 1E, Figure 3—figure supplement 1D and Figure 4—figure supplement 2C). Hence, even though IPN neurons are heterogenous, the various types of neurons are probably distributed throughout the IPN, and they were sampled in a similar way in our recordings. Therefore, in our point of view, further addressing neuronal diversity is not needed in the current study. In addition, the fact that our results are robust (e.g. 80% decrease in current in brain slices) without having to address neuronal diversity of the IPN further demonstrate the robustness of our findings.

2. The authors used a pseudo-ternary plot to differentiate nicotine "avoiders" and "non-avoiders". The main behavioral and electrophysiological factors that differentiate "avoiders" from "non-avoiders" can be but were not fully drawn, from the data presented. Also, there is still apparent heterogeneity between "avoiders" and "non-avoiders" (Figure 2A). How these differences relate to IPN nicotine-evoked currents, and if the power of the plot is sufficient for dissecting the population, can be further explored.

We classified mice in two groups (avoiders and non-avoiders) and recognized, in the original version of the paper, that there was heterogeneity within each group (Figure 2A). For this reason, we illustrated the diversity in consumption profiles using ternary plots, in which all individual mice are represented. Avoiders and non-avoiders were distinguished on the basis of their consumption profile (Figure 1), not on the basis of the ternary plots (Figure 2). The analysis of consumption in terms of choice using ternary plots does not aim to identify sub-populations (hence, there is no power problem). Rather, it shows that the two groups were associated with different consumption patterns (Figure 2C-D) and different amplitudes of nicotine-evoked current (Figure 2E,F), which not only reinforces our first classification, but also allows us to establish correlations between these parameters. In the following figures (Figure 3-5), we make a more direct link between these variables. In particular, we manipulate nAChR currents (either by chronically exposing mice to nicotine or by using mutant mice in which nAChR expression is altered) and link these effects to drug use and choice patterns.

In terms of further linking behavioral heterogeneity to nicotine responses, we performed a new analysis (see Author response image 2). We looked for a correlation between current amplitude and distance from each apex (0% nicotine apex, 100% nicotine apex and side bias apex). Unfortunately, we did not find a statistically-significant correlation (except for side bias), probably because of low power (n too small). We thus prefer not to include these analyses in the manuscript.

4. The change of in vivo activity of the IPN in Î²4 KO or OE rescue should be documented.

We have now performed this experiment as requested. We virally re-expressed b4 (or GFP for controls) in the IPN of b4^-/-^ mice, and recorded the response of IPN neurons to i.v. injections of nicotine (30 ug/kg). We found that the responses to nicotine were restored (new Figure 5C). These results confirm what we initially observed at the whole-cell level in patch clamp experiments, and fully validate our viral rescue strategy.

5. The individual difference in sensitivity to the bitterness of nicotine should be excluded by using nicotine/quinine vs. quinine preference tests.

As requested by all three reviewers, we have now performed a two-bottle choice experiment to verify whether different sensitivities to the bitterness of the nicotine solution could explain the different sensitivities to the aversive properties of nicotine. Indeed, even though we used saccharine to mask the bitterness of the nicotine solution, we cannot fully exclude the possibility that the taste capacity of the mice could affect their nicotine consumption. Reviewers 1 and 2 suggested to perform nicotine/quinine versus quinine preference tests, but we were afraid that forcing mice to drink an aversive, quinine-containing solution might affect the total volume of liquid consumed per day, and also might create a “generalized conditioned aversion to drinking water – detrimental to overall health and a confounding factor” as pointed out by reviewer 3. Therefore, we designed the experiment a little differently.

In this two-bottle choice experiment, mice were first proposed a high concentration of nicotine (100 µg/ml) which has previously been shown to induce avoidance behavior in mice (Figure 3C). Then, mice were offered three increasing concentrations of quinine: 30, 100 and 300 µM. Quinine avoidance was dose dependent, as expected: it was moderate for 30 µM but almost absolute for 300 µM quinine. We then investigated whether nicotine and quinine avoidances were linked. We found no correlation between nicotine and quinine preference (new Figure: Figure 1- supplementary figure 1D). This new experiment strongly suggests that aversion to the drug is not directly tied to the sensitivity of mice to the bitter taste of nicotine.

Other results reinforce this conclusion. First, none of the b4^-/-^ mice (0/13) showed aversion to nicotine, whereas about half of the virally-rescued animals (8/17, b4 re-expressed in the IPN of b4^-/-^ mice) showed nicotine aversion, a proportion similar to the one observed in WT mice. This experiment makes a clear, direct link between the expression of b4 nAChRs in the IPN and aversion to the drug.

Furthermore, we also verified that the sensitivity of b4^-/-^ mice to bitterness is not different from that of WT mice (new Figure 4 —figure supplement 1B). This new result indicates that the reason why b4^-/-^ mice consume more nicotine than WT mice is not because they have a reduced sensitivity bitterness.

Together, these new experiments strongly suggests that interindividual differences in sensitivity to the

bitterness of nicotine play little role in nicotine consumption behavior in mice.

Reviewer #2 (Recommendations for the authors):– The authors need to provide the sex of the mice used in each experiment. If both sexes were used across the study, the authors should describe how and if the estrus cycle was monitored.

Information on sex (males only) was provided both in the abstract and in the Results section, but was omitted in the methods section. This is now corrected.

– The authors should describe the power analysis used to determine the sample sizes of the current data set. For example, it is unclear why a double number of mice/neurons were recorded in N-Av while the related s.e.m. is 50% lower (Figure 2) compared to the Av group.

We did not calculate power, notably because we had no a priori idea of the expected results (difference between the two groups and variances). Furthermore, the distribution is non-gaussian, which makes the power calculation much more complicated. In Figure 2E and F, we initially included mice that underwent a slightly different behavioral protocol (they stopped the 2BC at 100 µg/ml instead of 200 µg/ml). We realize that this was not sufficiently explained in the manuscript, quite misleading, and in fact not necessary. In this new version of the manuscript, we therefore included only the mice that underwent the full protocol (up to 200 µg/ml), that is 7 nonavoiders (52 neurons) and 7 avoiders (57 neurons). The number of mice and neurons between the two groups is now more even. The difference in current amplitude between the two groups still holds (p = 0.0027) and the correlation is even slightly better (R^2^ = 0.47, p = 0.004).

– The IPn is a heterogeneous brain structure. Also, the authors should further describe the location of the recorded neurons and the viral expression.

Indeed, reviewer is right, and this could impact the interpretation of the results. As described above, we now provide maps of the neurons recorded both ex-vivo (patch-clamp recordings) and in-vivo (juxtacellular recordings). Overall, we have recorded in largely overlapping areas in most of the experimental groups (new supplementary figures, see details above). In addition, we haven’t observed any strong link between the medio-lateral or dorsoventral coordinates and the response to nicotine in the IPN (ex and in vivo). Therefore, we are confident that the effects we observe under chronic nicotine and after deletion of b4 are not due to sampling variabilities.

– The Y-axis in supplement figure 1D doesn't align well.

Indeed, corrected.

Reviewer #3 (Recommendations for the authors):– The oral model is a major weakness. I suggest using an alternative model (and/or species) with more translational relevance. Relatedly, it will be important for this work to use concentrations and doses of nicotine that are relevant to human tobacco users.

Certainly, the oral nicotine consumption model has limitations and we discuss these more extensively in the revised manuscript. But these limitations do not disqualify the method and for some aspects, and in particular regarding consumption strategies, this test is much more interesting than self-administration. Finally, the doses of nicotine used are totally in line with a large body of literature, see details in Matta, S. G. et al. 2006. Guidelines on nicotine dose selection for in vivo research. Psychopharmacology 190, 269–319.

– The fact that there are a variety of identifiable types of neurons in the IPN is a very important issue because this work did not take this into account. Differences in nicotine receptor function, noted by the authors, could be simply from sampling variability rather than due to the action of chronic nicotine. A version of this paper that makes a strong impact on the field will need to link changes in nAChR currents to specific types of neurons in the IPN.

We did not take into account the variety of identifiable types of IPN neurons because it is not necessary at this level of analysis. The reviewer suggests that differences in nicotine receptor function could be simply from sampling variability rather than due to the action of chronic nicotine. We now provide maps of the recorded neurons (in slices and in vivo), as well as evidence that neurons were sampled similarly between the different experimental conditions. We think, and we thank the three reviewers for pointing that issue out, that our findings are much more convincing now and that showing that nicotinic transmission in the interpeduncular nucleus impact nicotine preference in a free choice test is important to understand basic mechanisms that underly response to nicotine.

– This paper had aspects that were not novel, which prevented the paper from making a strong impact on the field. The manuscript needs a limitations section that provides a measured assessment of the weaknesses of the paper.

Limitations and novelty of our study are discussed more extensively in the revised version of the manuscript.

[Editors’ note: what follows is the authors’ response to the second round of review.]

The manuscript has been improved but there are some minor points that need to be addressed. Reviewer #3 mentioned that beta4 functional responses tend to be up-regulated following contingent or non-contingent nicotine treatment in the MHb/IPN pathway in multiple species. Elaboration in the discussion of this point is recommended.Reviewer #3 (Recommendations for the authors):The present revision is improved over the previous version and has addressed most of the review concerns. A key finding remains at odds with other findings in the literature, though. The manuscript reports that chronic, passive exposure to nicotine results in down-regulation of nAChRs, presumably beta4-containing receptors (identification of beta4-containing was inferred, not demonstrated with pharmacology). The paper discusses that their results are contradictory to those of Arvin et al. and offer speculation that this is due to methodological differences. The finding of nAChR down-regulation following nicotine treatment is actually contradictory to findings reported in many papers, across several labs, and across several species/preparations. Consider the following regarding the medial habenula / IPN pathway (where beta4 receptors are enriched) and in cultured cells:1. Shih et al. (Mol Pharm 2015). Chronic nicotine treatment in mice (mini pumps) results in upregulation of nAChRs in medial habenula, a brain area with similar beta4-containing nAChRs.2. Banala et al. (Nature Methods 2018). MHb nAChR upregulation following chronic nicotine in mice (nicotine drinking water).3. Jin et al. (eNeuro, 2020). Upregulation of nAChRs in MHb following nicotine self-administration in rats, shown with ACh puff experiments.4. Tapia et al. (Neuropharm. 2022). Nicotine self-administration in rats results in upregulation of nAChRs in IPN – demonstrated with ACh puff experiments, not uncaging.5. Zhao-Shea et al. (Current Biology 2013). Beta4-containing receptors are upregulated (functionally and mRNA) in dorsal IPN in response to chronic nicotine treatment in mice (nicotine drinking water).6. Mazzo et al. (J. Neurosci. 2013). Alpha3beta4 receptors are upregulated by nicotine treatment in vitro.Thus, the finding of downregulation of nAChRs in the present manuscript is interesting. The paper would be strengthened if there was an improved discussion of this issue.

We would like to thank you and the reviewers for the positive feedback. We have added a paragraph in the Discussion section, to address the remaining comment of reviewer 3. Notably, we discuss now in greater details the issue that the functional downregulation we have observed contrasts with previous findings. We also have added a few references, as listed by reviewer 3. We agree that this additional paragraph strengthens the discussion.

Our added paragraph (line 549-570): “This observed functional downregulation of nAChR currents contrasts with previous findings. Chronic nicotine was shown to cause functional upregulation of nicotinic currents in MHb neurons of mice (Arvin et al., 2019; Banala et al., 2018; Pang et al., 2016; Shih et al., 2015; Tucker and Drenan, 2020) and rats (Tucker and Drenan, 2020), as well as in IPN neurons of mice (Zhao-Shea et al., 2013) and rats (Tapia et al., 2022). However, it should be noted that upregulation is highly cell-type as well as receptor-subtype dependent (Nashmi et al., 2007; Shih et al., 2015; Zhao-Shea et al., 2013). For instance, in the mouse IPN, functional upregulation was shown only in SST-positive neurons, which constitute a small fraction of IPN neurons, and no increase in b4 subunit expression was reported in SST-negative neurons (Shih et al., 2015). Moreover, in cell culture, it was found that b2-, but not b4-containing nAChRs were upregulated after nicotine treatment (Wang et al., 1998), which agrees with our findings. Upregulation is likely due to an increase in receptor number (Banala et al., 2018), yet it can be masked by receptor desensitization, which occurs during prolonged nicotine treatment. In some of the reports of nicotine-induced upregulation, currents were recorded only after nicotine withdrawal (Pang et al., 2016), leaving enough time for the receptors to recover from desensitization. Here, we recorded responses to nicotine in vivo, while nicotine was still present, and found that they were reduced in amplitude, which fits with the downregulation we observed in slices. Furthermore, our behavioral data in nicotine-treated WT and Chrnb4-/- mice are in complete agreement: both displayed reduced responses to nicotine (ex vivo and in vivo) in IPN neurons, as well as increased nicotine intake compared to naive, WT animals.”

We have also added one sentence in the Results section, to refer to the GENSAT data, and added the corresponding reference.